# Optogenetic-controlled immunotherapeutic designer cells for post-surgical cancer immunotherapy

Yuanhuan Yu [1,4], Xin Wu [1,4], Meiyan Wang [1,4], Wenjing Liu [1], Li Zhang [1], Ying Zhang [1], Zhilin Hu [2], Xuantong Zhou [1], Wenzheng Jiang [1], Qiang Zou [2], Fengfeng Cai [3] ✉ & Haifeng Ye [1] ✉

Surgical resection is the main treatment option for most solid tumors, yet cancer recurrence after surgical resection remains a significant challenge in cancer therapy. Recent advances in cancer immunotherapy are enabling radical cures for many tumor patients, but these technologies remain challenging to apply because of side effects related to uncontrollable immune system activation. Here, we develop far-red light-controlled immunomodulatory engineered cells (FLICs) that we load into a hydrogel scaffold, enabling the precise optogenetic control of cytokines release (IFN-β, TNF-α, and IL-12) upon illumination. Experiments with a B16F10 melanoma resection mouse model show that FLICs-loaded hydrogel implants placed at the surgical wound site achieve sustainable release of immunomodulatory cytokines, leading to prevention of tumor recurrence and increased animal survival. Moreover, the FLICs-loaded hydrogel implants elicit long-term immunological memory that prevents against tumor recurrence. Our findings illustrate that this optogenetic perioperative immunotherapy with FLICs-loaded hydrogel implants offers a safe treatment option for solid tumors based on activating host innate and adaptive immune systems to inhibit tumor recurrence after surgery. Beyond extending the optogenetics toolbox for immunotherapy, we envision that our optogenetic-controlled living cell factory platform could be deployed for other biomedical contexts requiring precision induction of biotherapeutic dosage.

Surgery is the principal therapeutic mode for malignant solid tumors. Despite continual improvements in surgical techniques, residual microtumors are difficult to remove completely and continue to cause tumor recurrence after resection[1]. Moreover, the resultant wound healing process and perioperative trauma-associated inflammation caused by surgery can promote tumor recurrence, which can result in significant morbidity and mortality by suppressing the activity of antitumor leukocytes including natural killer (NK) cells and cytotoxic T lymphocytes (CTL, CD8+ T cell)[2,3]. Accordingly, new strategies to prevent cancer recurrence after surgery are urgently sought.

[1]Shanghai Frontiers Science Center of Genome Editing and Cell Therapy, Biomedical Synthetic Biology Research Center, Shanghai Key Laboratory of Regulatory Biology, Institute of Biomedical Sciences and School of Life Sciences, East China Normal University, Dongchuan Road 500, Shanghai 200241, China. [2]Shanghai Institute of Immunology, Shanghai Jiao Tong University School of Medicine, 280 South Chongqing Road, Shanghai 200025, China. [3]Department of Breast Surgery, Yangpu Hospital, School of Medicine, Tongji University, Shanghai 200090, China. [4]These authors contributed equally: Yuanhuan Yu, Xin Wu, Meiyan Wang. ✉e-mail: caifengfeng@tongji.edu.cn; hfye@bio.ecnu.edu.cn

Immunotherapies are revolutionizing oncology, and there is interest in combining immunotherapy with surgical resection as an approach to quickly remove tumors while also helping to prevent postoperative recurrence[4–6]. A variety of drugs and cell-based technologies can now be used to boost the body's natural immune system to fight cancer and produce tumor-specific memory T cells to achieve long-term systemic immunosurveillance that can potentially protect against post-resection recurrence[7–9]. However, the uncontrolled delivery and release of immunotherapeutics can cause excessive immune responses that can lead to fatal side effects[10,11].

Seeking to address this, several studies have worked with mouse tumor resection models and demonstrated efficacy of biomaterials-based local delivery systems that can prolong the release of diverse immune agonists (e.g., TLR7/8, STING) or antibodies (e.g., anti-CD47), including for example a biodegradable hydrogel encasing these agents placed at the tumor resection site[12,13]. There have also been studies showing that scaffolds can be used to control the delivery of cells, including both tumor-reactive T cells[14,15] and mesenchymal stem cells expressing an immune agonist (IFN-β)[16]. Very recently, a hydrogel scaffold containing chimeric antigen receptor T cells (CAR-T cells) and platelets conjugated with the checkpoint inhibitor programmed death-ligand 1 (PDL1) was shown to inhibit both local tumor recurrence and the growth of distant tumors[17]. However, it is notable that none of these technologies have yet exploited the unique biological responsivity of cells to enable precise spatiotemporal control of these immunotherapeutic interventions.

The complementary field of synthetic designer cell-based therapeutics has shown that cells engineered to be responsive to physical and chemical signals can precisely regulate the sustainable release of protein drugs to treat various diseases[18–21] including cancer[22,23]. And there is particular enthusiasm for optogenetics-based cell responsivity because using light can achieve remote, traceless, well-defined modulation of cellular activities in a non-invasive way compared with chemical signals. Our research group previously developed an optogenetic tool based on far-red light (FRL) inducible genetic modules that has a capacity for deep tissue penetration[24], so we envisioned that it may be possible to combine designer living cell factories with FRL induction as a concept for the precise (and thus safe) release of immunomodulatory cytokines as a way to help protect against tumor recurrence post-resection.

Here, we develop far-red light-controlled immunomodulatory engineered cells (FLICs) and then embed them within a hydrogel scaffold for implantation at resection sites. This system enables precise, FRL-illumination-induced spatiotemporal control of the production and release of multiple immunotherapeutic cytokines (IFN-β, TNF-α, and IL-12)[25–28]. After confirming that FLICs can robustly and tunably induce production upon FRL illumination, we deliver our FLICs-loaded hydrogel implants to tumor sites after surgical resection of melanoma tumors. We find that FLICs-loaded hydrogel implants confer external control of the location and timing of IFN-β, TNF-α, and IL-12 expression upon illumination with a non-invasive FRL LED light, and confirm that this activated both the innate and adaptive immune systems, thereby preventing tumor recurrence and substantially outperforming soluble immunotherapeutic cytokine administration. We also demonstrate that FLICs-loaded hydrogel implants elicit a long-term immune memory response that inhibit distant tumor recurrence.

## Results

### Characterization of the cytokine secretion from FLICs-loaded hydrogel implants

To develop our optogenetic control cytokine production system for preventing tumor recurrence after surgical resection, we constructed FRL-controlled immunomodulatory engineered cells (FLICs) based on our previously reported orthogonal FRL-triggered optogenetic system (FRL-v2), which comprises an engineered bacterial FRL-activated cyclic diguanylate monophosphate (c-di-GMP) synthase (BphS, from *Erythrobacter litoralis*) and a c-di-GMP-responsive hybrid transactivator, p65-VP64-BldD. Under FRL illumination (730 nm, from LEDs), intracellular guanylate triphosphate (GTP) can be converted into c-di-GMP by BphS. c-di-GMP can trigger homodimerization of the mammalian FRL-dependent hybrid transactivator (p65-VP64-BldD) to promote binding to the FRL-v2-specific chimeric promoter ($P_{FRL}$) to induce transcription of genes for cytokines including murine IFN-β, TNF-α, and IL-12 (Fig. 1).

Subsequently, we transfected human mesenchymal stem cells (hMSC-TERT) with plasmids carrying the light receptor BphS and the FRL-dependent transactivator (pYH88, p65-VP64-BldD), and the immunomodulatory cytokine genes driven by the light responsive promoter ($P_{FRL}$) (pYH428) (Supplementary Fig. 1a). After confirming the light-induced cytokine release (IFN-β, TNF-α, and IL-12) after illumination (Supplementary Fig. 1b–e), then we screened various FLICs by enhanced green fluorescent protein (EGFP) expression and evaluated the production profile of released IFN-β, TNF-α, and IL-12 of each cell lines, and we found that clone no. 16 displayed high cytokine production profile with more than 200-fold induction (Supplementary Fig. 2a–d). In vitro testing established that the FLICs produce the three cytokines in a FRL illumination intensity- and exposure-time dependent manner (Supplementary Fig. 3a–c and d–f). Moreover, the system displayed fully reversible cytokine expression kinetics (Supplementary Fig. 3g–i). This result indicates that the FRL-triggered cytokine production from FLICs is tunable.

To protect the cells from the host's immune system, while simultaneously allowing free diffusion of oxygen, ions, and secreted proteins, we then encapsuled the FLICs in a commercially available polysaccharide-based biocompatible hydrogel scaffolds and studied cytokine production profile by these FLICs-loaded hydrogel implants (Fig. 2a, b). Assays of the FLICs-loaded hydrogel implants immersed in media showed that encapsulation of the FLICs into the hydrogel scaffold does not disrupt the expected stable, long-term, illumination intensity- (Fig. 2c–e) and exposure-time (Fig. 2f–h) dependent production of IFN-β, TNF-α, and IL-12. Moreover, cytokine production from the FLICs-loaded hydrogel implants was fully reversible: almost no IFN-β, TNF-α, or IL-12 was detected within 24 h after stopping illumination (Fig. 2i–k). Finally, testing with 4 h of FRL illumination per day for 30 days confirmed that the FLICs are capable of stable, long-term optogenetically induced cytokine production (Supplementary Fig. 4a–c). This result indicates that FRL-triggered cytokine production from FLICs-loaded hydrogel implants is also robust and tunable.

### Immunotherapeutic FLICs-loaded hydrogel implants for preventing tumor recurrence after surgical tumor resection

Having confirmed that the release of these immunotherapeutic cytokines mediated by FLICs-loaded hydrogel implants can be controlled precisely in vitro, we first evaluated the stability and degradability of the hydrogel implants and found that these implants could stay in mice for at least 8 weeks without observed degradation (Supplementary Fig. 5). We next assessed the utility of the FLICs-loaded hydrogel implants in mouse tumor resection models. Female C57BL/6J mice were inoculated with B16F10_Luc melanoma cells (subcutaneous, dorsum). On day 7 after inoculation, the mice were imaged by bioluminescent IVIS imaging, which confirmed that the size of the tumors (based on the bioluminescence signals of B16F10_Luc cells) was consistent across animals and enabled randomization into groups. After IVIS imaging, the tumors (80–100 mm³) were resected and then hydrogel scaffolds were placed in the tumor resection site; we tested implants harboring different cells: FLICs, wild-type hMSC-TERT, and hMSC-TERT constitutively expressing the cytokines. We also included an empty hydrogel scaffold control (without any cells), a hydrogel scaffold loaded with the cytokines, and included control mice which were implanted with hydrogel implants containing FLICs but were not

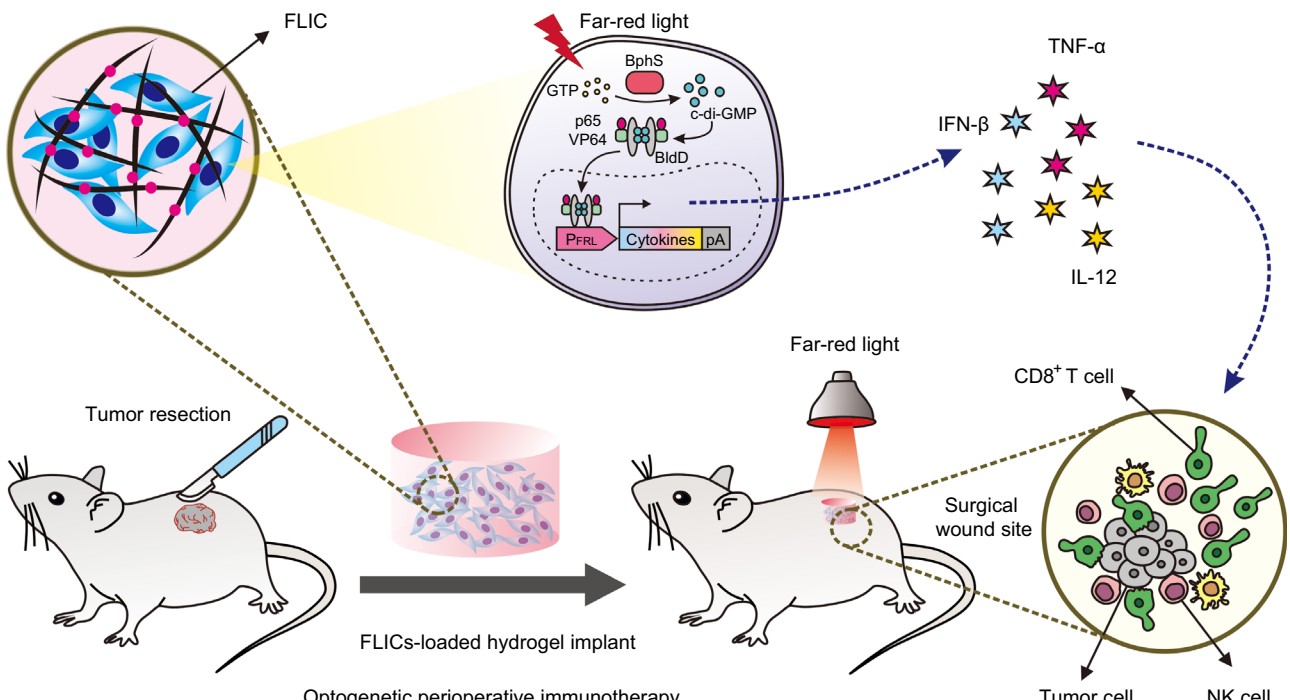

**Fig. 1 | Schematic showing the mouse experimental design and procedure for optogenetic perioperative immunotherapy mediated by FLICs-loaded hydrogel implants.** Far-red light-controlled immunomodulatory engineered cells (FLICs) were encapsulated within a polysaccharide-based biocompatible hydrogel that allows free diffusion of oxygen, ions, and secreted proteins while simultaneously shielding the encapsulated cells from the host immune system. After surgical resection of the B16F10 melanoma tumors in mice, the encapsulated FILCs were implanted subcutaneously at the surgical wound site. An engineered FRL-activated cyclic diguanylate monophosphate (c-di-GMP) synthase (BphS) from a bacterium converts intracellular guanylate triphosphate (GTP) into cyclic diguanylate monophosphate (c-di-GMP), which binds to the hybrid transactivator p65-VP64-BldD, thereby facilitating its translocation into the nucleus, where the transactivator binds its chimeric promoter $P_{FRL}$ to induce transcription of genes for mouse immunomodulatory cytokines (IFN-β, TNF-α, and IL-12). These cytokines promote activation of innate immunity, including antitumor NK cell and T cell responses to prevent tumor relapse.

given FRL illumination. The growth of recurrent tumors was monitored on post-resection days 3, 10, 17, and 24 (based on the bioluminescence signals of B16F10$_{Luc}$ cells), and mouse survival was assessed at the pre-determined end-point of 60 days (Fig. 3a).

The hydrogel implants loaded with FLICs (with FRL illumination) conferred strong inhibition against tumor recurrence (Fig. 3b–f and Supplementary Fig. 6). The tumor sizes in mice correlated with their survival: FLICs-loaded hydrogel implants (with FRL illumination) significantly prolonged mouse survival with a 100% survival rate after 60 days compared with other control groups. We also compared FLICs-loaded hydrogel implants with the hydrogel implants loaded with the three cytokines (1000 ng of IFN-β, 60 ng of TNF-α, 200 ng of IL-12 per hydrogel implant), and direct subcutaneous injection of the recombinant cytokine proteins to the resected tumor region (200 ng of IFN-β, 12 ng of TNF-α, 40 ng of IL-12 per mouse, once every three days over 15 days, five times in total). The FLICs-loaded hydrogel implants (with FRL illumination) were superior to the hydrogel implants loaded with the three cytokines and injection of the three recombinant cytokine proteins in terms of recurrent tumor volume. Together, these results establish that the FLICs-loaded hydrogel implants (under FRL illumination) confer protection against tumor recurrence.

To evaluate the potential toxicity of the FLICs-loaded hydrogel implants, fourteen days after tumor resection and placement of FLICs-loaded hydrogel implants, whole blood was collected and subjected to routine blood test [including white blood cells (WBC), red blood cells (RBC), hemoglobin (HGB), hematocrit (HCT), mean corpuscular volume (MCV), mean corpuscular hemoglobin (MCH), mean corpuscular hemoglobin concentration (MCHC), platelets (PLT), platelet crit (PCT)] (Supplementary Fig. 7a–i) and serum biochemistry analyses [including liver enzymes alanine aminotransferase (ALT), aspartate

aminotransferase (AST), and surrogate of kidney function blood urea nitrogen (BUN)] (Supplementary Fig. 7j–l). No obvious differences in these blood phenotypes were detected between controls (untreated healthy mice) and animals given implants and FRL illumination (Supplementary Fig. 7a–l). These results indicate that the placement of the FLICs-loaded hydrogel implants at resected tumor sites does not induce any obvious systemic adverse effects.

## Immunotherapeutic FLICs-loaded hydrogel implants mediated anti-tumor immune responses

We repeated the tumor inoculation and resection procedure described above with 5 groups: FLICs-loaded hydrogel implants with or without FRL illumination, hydrogel implants loaded with cells constitutively expressing cytokines or implants loaded with the three cytokines, and control group only resection but lacking any implant. Peripheral blood (days 7 and 14) and spleens (days 7 and 14) were collected and analyzed to monitor changes in immune related cytokine levels and in immune cell populations including innate and adaptive immune systems. Multiplexing laser bead-based immunoassay of peripheral blood samples showed that mice given the FLICs-loaded hydrogel implants and FRL illumination had increased levels of IFN-β, TNF-α, IL-12, CXCL10, IFN-α, and IFN-γ (Supplementary Fig. 8a–r), and tunable cytokines (IFN-β, TNF-α, IL-12) production from FLICs (Supplementary Fig. 9a–d). Given previous reports showed that cytokines regulate both innate and adaptive immune responses in autocrine and paracrine manners throughout the host immune system. It seems likely that the increased CXCL10, IFN-α, and IFN-γ levels result from the elevated IFN-β, TNF-α, and IL-12 accumulation in these mice. Of note, mice given the hydrogel implants loaded with hMSC-TERT constitutively expressing cytokines showed the highest IL-6 expression levels probably due to

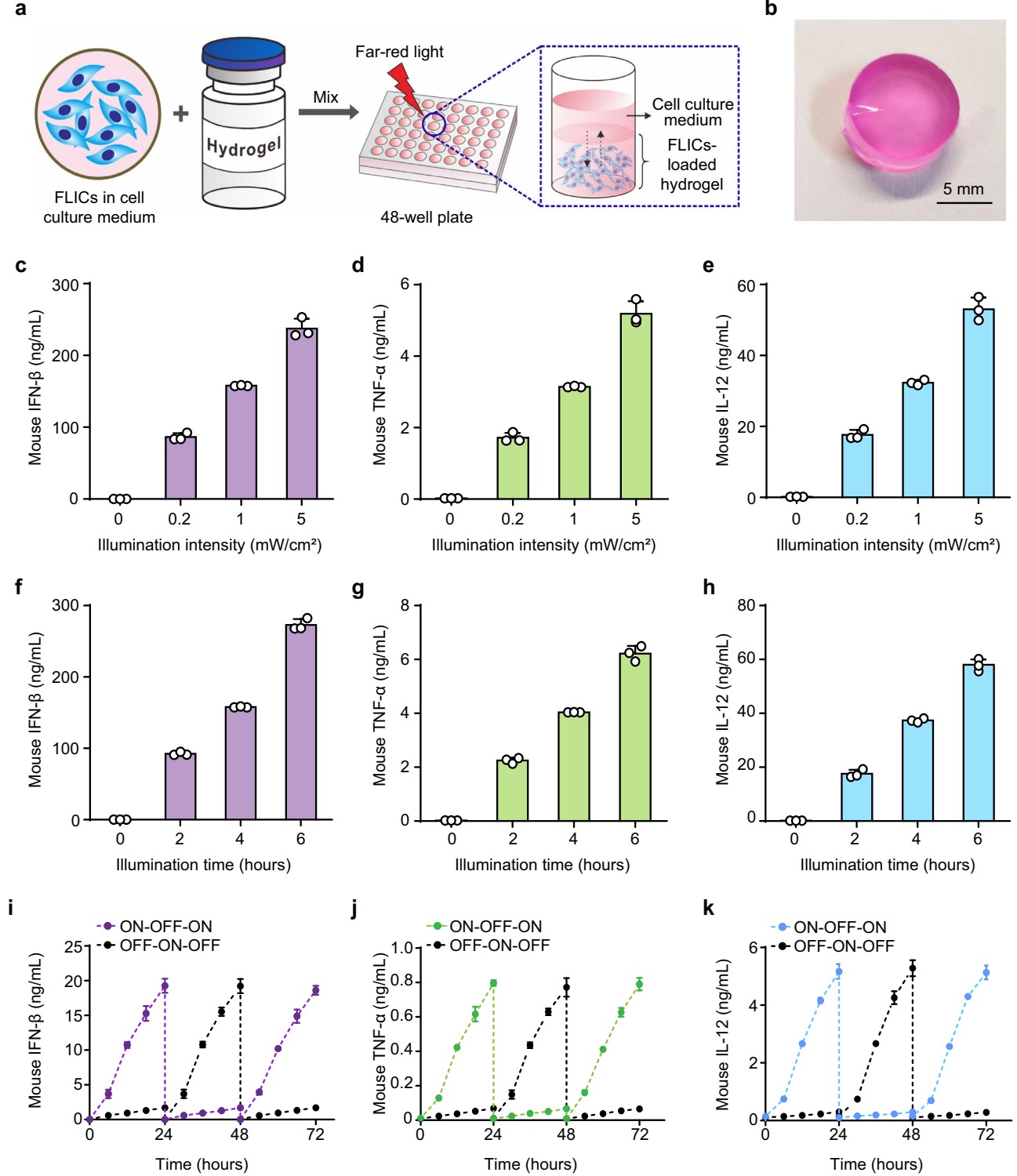

the overstimulated immune response among all the groups (Supplementary Fig. 10a–d). Collectively, these findings support the observation that FLICs-loaded hydrogel implants with illumination can trigger the expected IFN-β, TNF-α, and IL-12 secretion at the therapeutic concentrations without obvious toxic side effects.

To dissect the changes in immune cell population, we then recovered spleens and peripheral blood from sacrificed mice and analyzed spleens lymphocytes as well as peripheral blood lymphocytes via flow cytometry. The ratio of NK cells (CD3⁻NK1.1⁺) among total lymphocytes (CD45⁺ cell population) and activated (CD69⁺ and CD11b⁺CD27⁺) NK cells gating on CD3⁻NK1.1⁺ cells was significantly increased after treatment with FLICs-loaded hydrogel implants under FRL illumination in the spleen and peripheral blood of mice (Fig. 4a–f and Supplementary Fig. 11a–f). In addition, FLICs-loaded hydrogel implants under FRL illumination induced a much higher percentage of total CD8⁺ T cells among total CD45⁺CD3⁺ cell population and activated (CD69⁺) CD8⁺ T cells gating on total CD8⁺ cells compared with the various control groups (Fig. 4g–j and Supplementary Fig. 11g–j). Next, we measured IFN-γ production in CD8⁺ T cells following PMA and ionomycin stimulation for 5 h. We observed that mice given the FLICs-

**Fig. 2 | Characterization of cytokine production kinetics mediated by FLICs-loaded hydrogel implants. a** Schematic of the experimental procedure for preparation of FLICs-loaded hydrogel implants. FLICs ($2.5 \times 10^6$) in 60 μL Dulbecco's Modified Eagle Medium were mixed with 240 μL polysaccharide-based biocompatible hydrogel solution. Hydrogel matrix is crosslinked by ions (such as $Ca^{2+}$ or $Na^+$) interactions from the cell culture medium and then solidified in a 48-well plate. **b** Photograph showing a representative FLICs-loaded hydrogel implant. **c–e** Illumination-intensity-dependent cytokines (IFN-β, TNF-α, and IL-12) release kinetics of the FLICs-loaded hydrogel implants. FLIC ($2.5 \times 10^6$)-loaded hydrogel implants were illuminated with FRL at different light intensities (0, 0.2, 1, 5 mW/$cm^2$) for 4 h once a day for 2 days, and culture supernatants were collected to quantify (**c**) IFN-β, (**d**) TNF-α, and (**e**) IL-12 production using enzyme-linked immunosorbent assay (ELISA) kits 48 h after the first illumination.

**f–h** Exposure-time-dependent cytokines (IFN-β, TNF-α, and IL-12) release kinetics of the FLICs-loaded hydrogel implants. FLIC ($2.5 \times 10^6$)-loaded hydrogel implants were illuminated with FRL (1 mW/$cm^2$; 730 nm) for 0, 2, 4, 6 h once a day for 2 days, culture supernatants were collected to quantify (**f**) IFN-β, (**g**) TNF-α and (**h**) IL-12 production using corresponding ELISA kits 48 h after the first illumination. **i–k** Reversibility of cytokines (IFN-β, TNF-α, and IL-12) release mediated by the FLICs-loaded hydrogel implants. FLICs-loaded hydrogel implants were either illuminated with FRL (1 mW/$cm^2$) for 20 min (ON) or instead kept in the dark (OFF), and cytokines (**i**) IFN-β, (**j**) TNF-α, and (**k**) IL-12 production was scored every 6 h for 72 h. At the beginning of each 24-h cycle, the culture medium was replaced with fresh medium. Data in **c** to **k** are presented as the mean ± SD; *n* = 3 independent experiments. Source data are provided as a Source Data file.

loaded hydrogel implants and FRL illumination had increased the IFN-γ⁺CD8⁺ T cells gating on total CD8⁺ T cells compared with the various control groups (Fig. 4k). Subsequently, we evaluated the direct impact of NK cells and CD8⁺ T cells on inhibiting tumor growth, and we depleted each of these cell types using neutralizing antibodies. Depletion of NK cells and CD8⁺ T cells significantly attenuated the antitumor effects in FLICs-loaded hydrogel implants (with FRL) group (Fig. 4l, m). These data suggest that the FLICs-loaded hydrogel implants promoted tumor inhibition in a manner primarily dependent on NK cells and CD8⁺ T cells. These results confirmed that FLICs-loaded hydrogel implants elicit innate immune activation effects through priming NK cells, and adaptive immune anti-tumor responses through CD8⁺ T cells.

In addition, the antigen-specific T cell responses elicited by FLICs-loaded hydrogel implants in vivo were investigated. Female C57BL/6J mice were inoculated with B16F10 ovalbumin (OVA) expressing melanoma cells followed by resection of tumors and FLICs-loaded hydrogel implants were placed in the tumor resection site. On post-resection day 7, T cell response was determined by ex vivo re-stimulation of splenocytes with OVA$_{257-264}$ peptide in vitro and measurement of IFN-γ⁺CD8⁺ T cells. As shown in Supplementary Fig. 12, FLICs-loaded hydrogel implants under FRL illumination induced higher proportions of OVA$_{257-264}$ peptide-specific IFN-γ⁺CD8⁺ T cells from the total CD8⁺ T cell population compared with other control groups. These results indicate that FLICs-loaded hydrogel implants can induce antigen-specific CD8⁺ T cell responses and antitumor immunity.

We also conducted experiments wherein B16F10$_{Luc}$ melanoma cells were subcutaneously inoculated into both sides of the mouse dorsum, followed by resection of tumors on both sides, but implantation of FLICs-loaded hydrogel implants only on the right side. The growth of recurrent tumors was monitored on post-resection days 3, 10, 20, and 30 (based on the bioluminescence signals of B16F10$_{Luc}$ cells as before), and mouse survival was assessed at the pre-determined end-point of 60 days (Fig. 5a). Compared to control animals (not given any implant, or without FRL) we found that local tumor recurrence (i.e., on the right side) was significantly inhibited by FLICs-loaded hydrogel implants (with FRL), and also observed that the extent of tumor recurrence was significantly reduced on the left side (Fig. 5b–d). Again, mice of the FLICs-loaded hydrogel implants (with FRL) group had higher survival rates than the controls (Fig. 5e). Consistent with these results, higher infiltration of CD8⁺ T cells were observed in the distant tumor of FLICs-loaded hydrogel implants treated mice with corresponding decreased tumor size compared to the untreated control and without FRL groups (Supplementary Fig. 13). Thus, FLICs-loaded hydrogel implants can confer protection against the recurrence of both local and distant tumors.

### FLICs-loaded hydrogel implants for T cell memory response inducing durable immune responses

Evidence has accumulated that these three cytokines participate in the differentiation of naive CD8⁺ T cells to memory CD8⁺ T cells

development and survival[29,30]. We were thus interested to determine whether the FLICs-loaded hydrogel implants can induce memory immune responses (Fig. 6a). Indeed, analysis of both peripheral blood and spleens on day 14 after tumor resection and implantation showed significantly elevated proportions of (CD44⁺CD62L⁺) CD8⁺ central memory T cells among total CD8⁺ T cell population in the FLICs-loaded hydrogel implants (with FRL) group compared to controls (Fig. 6b, c).

We then designed a tumor re-challenge experiment in which the mice were injected with $1 \times 10^5$ B16F10$_{Luc}$ melanoma cells at the original tumor site on post-resection day 14 followed by analysis of tumor growth (IVIS bioluminescence) and mouse survival. For naive mice, tumor growth was clearly detected on day 11 after the tumor cell re-challenge, and their tumors continued to grow larger throughout the observation period. In contrast, mice of the FLICs-loaded hydrogel implant (with FRL) group did not show obvious tumor growth after re-challenge (Fig. 6d–f). We also found that the overall survival of the FLICs-loaded hydrogel implant (with FRL) animals was significantly longer than the control group (Fig. 6g). These results indicate that post-resection installation our FLICs-loaded hydrogel implants induces T-cell memory responses that confer extended immunity against tumor recurrence as expected.

## Discussion

Cancer recurrence after surgical resection remains a significant cause of treatment failure. Ultimately seeking to help improve cure rates after tumor resection and to minimize the risk of excessive T cell activation, we here developed far-red light-controlled immunomodulatory engineered cells (FLICs) and loaded them into a hydrogel scaffold. We achieved optogenetic control production of immunomodulatory cytokines (IFN-β, TNF-α and IL-12), which conferred strong protection against post resection tumor recurrence and improved long-term survival rates compared with the cytokine loaded implant systems. Importantly, our data show that this technology supports rapid cessation of immunomodulatory cytokine release, avoiding overstimulation of the immune response and we experimentally confirmed that the FLICs-loaded hydrogel implants placed in the tumor resection site activated both innate and adaptive immune responses. Our results showcase strong inhibition of distant tumor growth, without any obvious toxicity, and while inducing durable antitumor T cell memory.

There is now extensive support for the idea that cytokine-based immunotherapies can induce the establishment of favorable conditions for the host immune system postresection; and this is viewed as an excellent approach for ongoing tumor elimination[31,32]. Injection of cytokines will offer a more practical route of administration. However, a single-dose monotherapy of a cytokine often confers no obvious therapeutic benefit[33–35], while high-dose treatment can lead to side effects reflecting excessive activation of the immune system[16,36]. It is now clear that understanding of dose scheduling—and our ability to control the dosage of complex biological interventions—needs to be

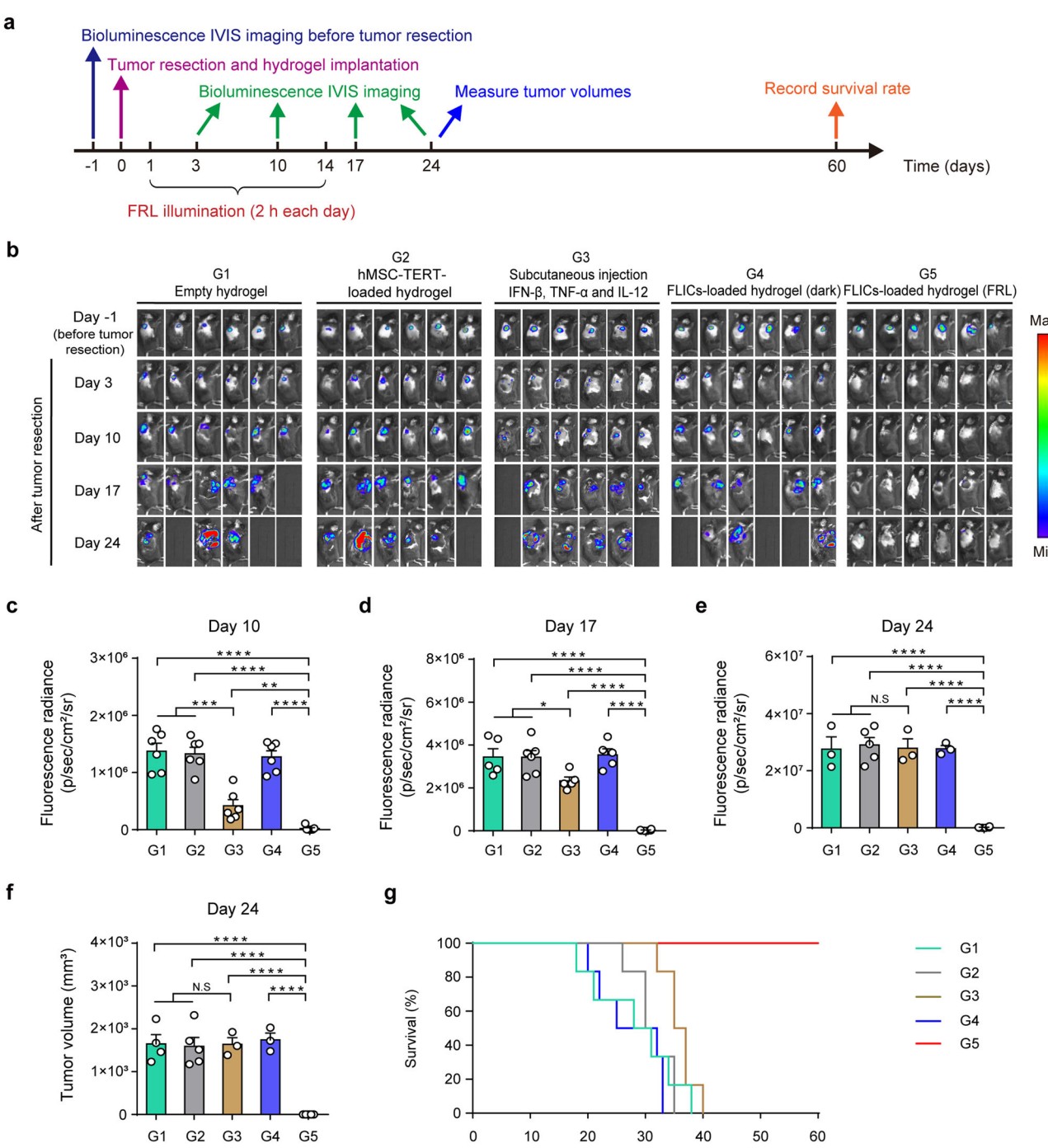

**Fig. 3 | Optogenetic control of cytokine release from FLICs-loaded hydrogel implants prevents tumor recurrence after surgical tumor resection of B16F10 melanoma tumors in C57BL/6J mice. a** Schematic illustrating the experimental procedure and the time schedule used for evaluating FLICs-loaded hydrogel implant performance. **b** Serial in vivo bioluminescence imaging of B16F10$_{Luc}$ tumors expressing luciferase before surgery and after surgical resection of primary tumors following resection site implantation of hydrogel scaffolds loaded with FLICs (IFN-β, TNF-α, and IL-12), or empty hydrogel scaffolds, or wild type hMSC-TERT, or local injection of the recombinant cytokine proteins (200 ng of IFN-β, 12 ng of TNF-α, 40 ng of IL-12 per mouse, once every three days over 15 days, five times in total). The mice were illuminated with FRL (10 mW/cm²; 730 nm) for 2 h each day for 14 days. Mice implanted with wild-type hMSC-TERT or empty hydrogels, or mice bearing FLICs-loaded hydrogel implants but not exposed to FRL

illumination were examined as controls. Six mice per group are shown. Quantification of tumor burden based on bioluminescence IVIS imaging shown in **b** on day 10 (**c**), day 17 (**d**), and day 24 (**e**) after implantation. **c** ***$P$ = 0.0003 for G1 versus G3; ***$P$ = 0.0002 for G2 versus G3; **$P$ = 0.0045 for G3 versus G5; ****$P$ < 0.0001 for all other groups versus G5. **d** *$P$ = 0.0304 for G1 versus G3; *$P$ = 0.0181 for G2 versus G3; ***$P$ < 0.0001 for all other groups versus G5. **e** ****$P$ < 0.0001 for all other groups versus G5. **f** Tumor volume measurement in different mouse groups 24 days after tumor surgical resection. ****$P$ < 0.0001 for all other groups versus G5. **g** Kaplan-Meier curves for mouse survival ($n$ = 6 mice). Data in **c** to **f** are presented as the mean ± SEM ($n$ = 6 mice). $P$ values were calculated by two-tailed unpaired $t$-test. *$P$ < 0.05, **$P$ < 0.01, ***$P$ < 0.001, ****$P$ < 0.0001. N.S, no significance. Source data are provided as a Source Data file.

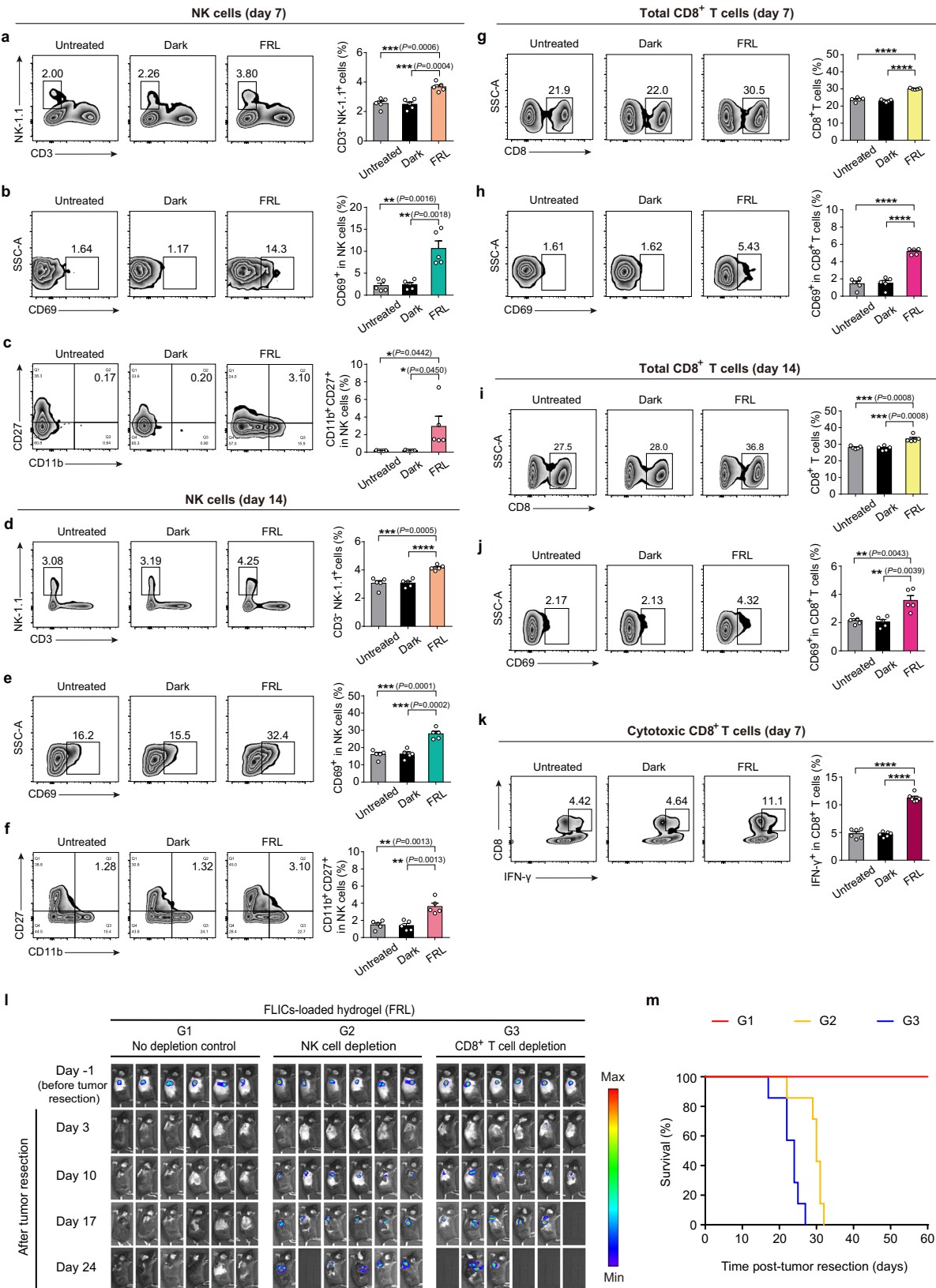

deepened before researchers can design clinical trials likely to deliver on the strong preliminary promise cytokine immunotherapeutics. In this context, the capacity of our FLICs-loaded hydrogel implants to efficiently and non-invasively control immunotherapeutic cytokine production based on controlling light irradiation times and intensities should be particularly attractive. Indeed, our findings suggest

that the type of optogenetic immunotherapy we have illustrated can open the door for further clinical development of safe cytokine immunotherapies.

Despite the strong inhibition of tumor growth using the optogenetic immunotherapy, there is still room for improvement with regard to the convenience of this therapy. A long illumination time is

**Fig. 4 | Anti-tumor immune response mediated by FLICs-loaded hydrogel implants. a–c** The activated and effector phenotypes of NK cells evaluated in spleen of mice sacrificed on day 7 by flow cytometry. Representative flow cytometric analysis images (left) and corresponding quantification (right) of **a** NK cells, **b** CD69+ NK cells, and **c** CD11b+CD27+ NK cells. Representative flow cytometric analysis images (left) and corresponding quantification (right) of (**d**) NK cells, (**e**) CD69+ NK cells, and (**f**) CD11b+CD27+ NK cells in spleens of mice sacrificed on day 14. Representative flow cytometric analysis images (left) and corresponding quantification (right) of **g** Total CD8+ T cells and **h** CD69+CD8+ T cells in spleens of mice sacrificed on day 7. Representative flow cytometric analysis images (left) and corresponding quantification (right) of (**i**) Total CD8+ T cells and (**j**) CD69+CD8+ T cells in spleens of mice sacrificed on day 14. **k** The percentage of IFN-γ+CD8+ T cells in lymphocytes from spleen of mice sacrificed on day 7. Splenocytes were cultured

with phorbol myristate acetate (PMA, 50 µg/mL) and ionomycin (50 µg/mL) at 37 °C. After 12 h, IFN-γ+CD8+ T cells were analyzed by intracellular cytokine staining flow cytometry. Representative flow cytometric analysis images (left) and corresponding quantification (right) of IFN-γ+CD8+ T cells. **l** Serial in vivo bioluminescence imaging of B16F10_{Luc} tumors expressing luciferase before surgery and after surgical resection of primary tumors following resection site implantation of hydrogel scaffolds loaded with FLICs (IFN-β, TNF-α, and IL-12) with or without intraperitoneal injection of anti-mouse NK-1.1 or anti-mouse CD8α. Six mice per group are shown. **m** Kaplan-Meier curves for mouse survival ($n = 7$ mice). Data in **a**–**k** are presented as the mean ± SEM (**a**–**j**, $n = 5$ mice; **k**, $n = 6$ mice) except representative flow cytometric analysis images. *P* values were calculated by two-tailed unpaired *t*-test. *$P < 0.05$, **$P < 0.01$, ***$P < 0.001$, ****$P < 0.0001$. Source data are provided as a Source Data file.

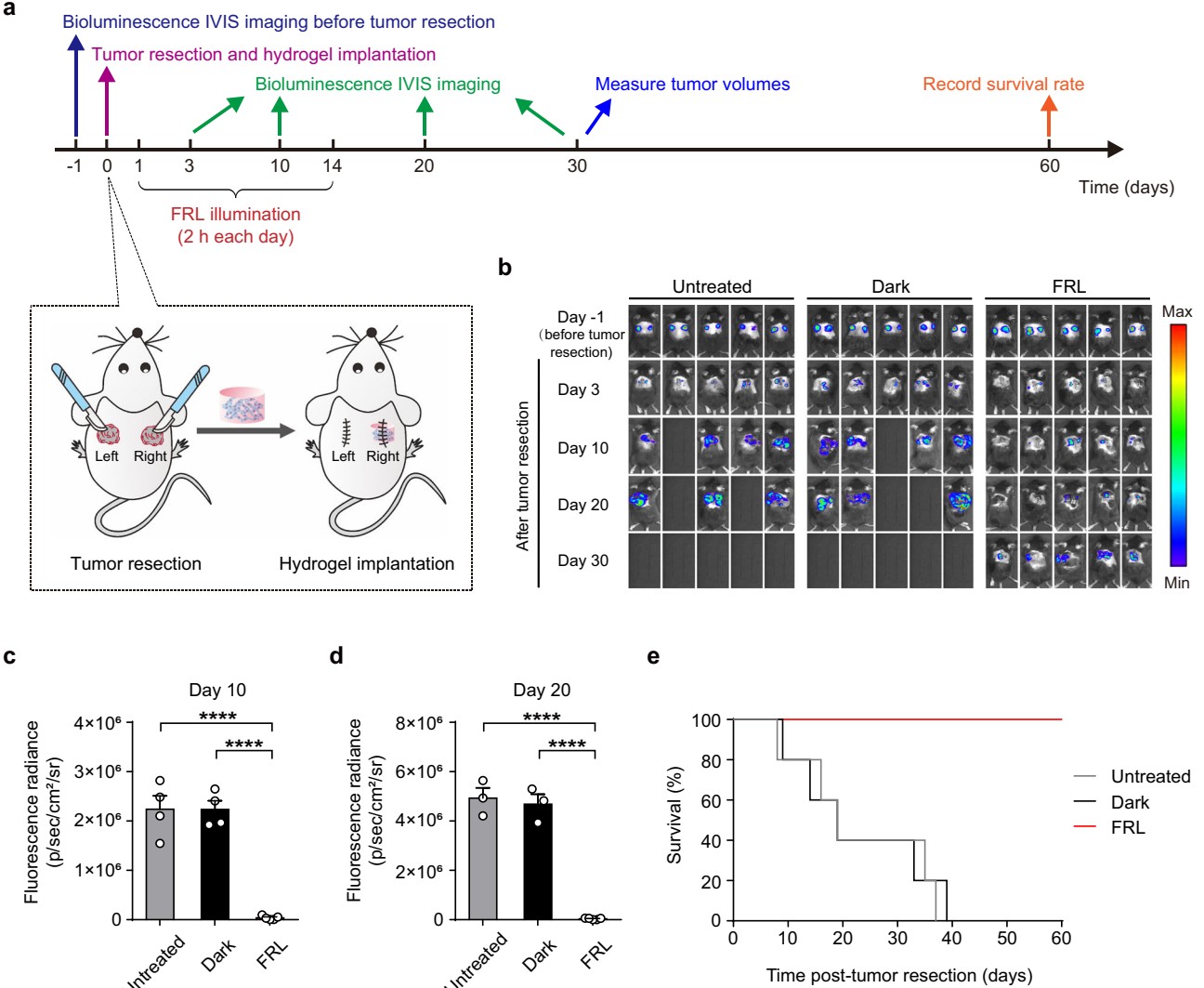

**Fig. 5 | Systemic anti-tumor immune response mediated by FLICs-loaded hydrogel implants in C57BL/6J mice. a** Schematic illustrating the experimental procedure and the time schedule for evaluating systemic anti-tumor immune response mediated by FLICs-loaded hydrogel implant in B16F10 melanoma resection mouse model. Mice were subcutaneously (s.c.) inoculated with B16F10_{Luc} cells in the right and left flanks. Treated mice were implanted with FLICs-loaded hydrogel implants only on the right flank, and mice were then illuminated with FRL (10 mW/cm²; 730 nm) for 2 h each day for 14 days. Control groups included mice

not given FRL illumination and mice that were not given a post-resection implant (untreated). **b** In vivo bioluminescence imaging of B16F10_{Luc} tumor burden on both sides before and after tumor resection. Five mice per group are shown. Quantification of tumor burden based on the bioluminescence IVIS imaging from **c** on day 10 (**c**) and on day 20 (**d**). Data in **c** and **d** are presented as the mean ± SEM ($n = 5$ mice). *P* values were calculated by two-tailed unpaired *t*-test. ****$P < 0.0001$. **e** Kaplan-Meier curves for mouse survival ($n = 5$ mice). Source data are provided as a Source Data file.

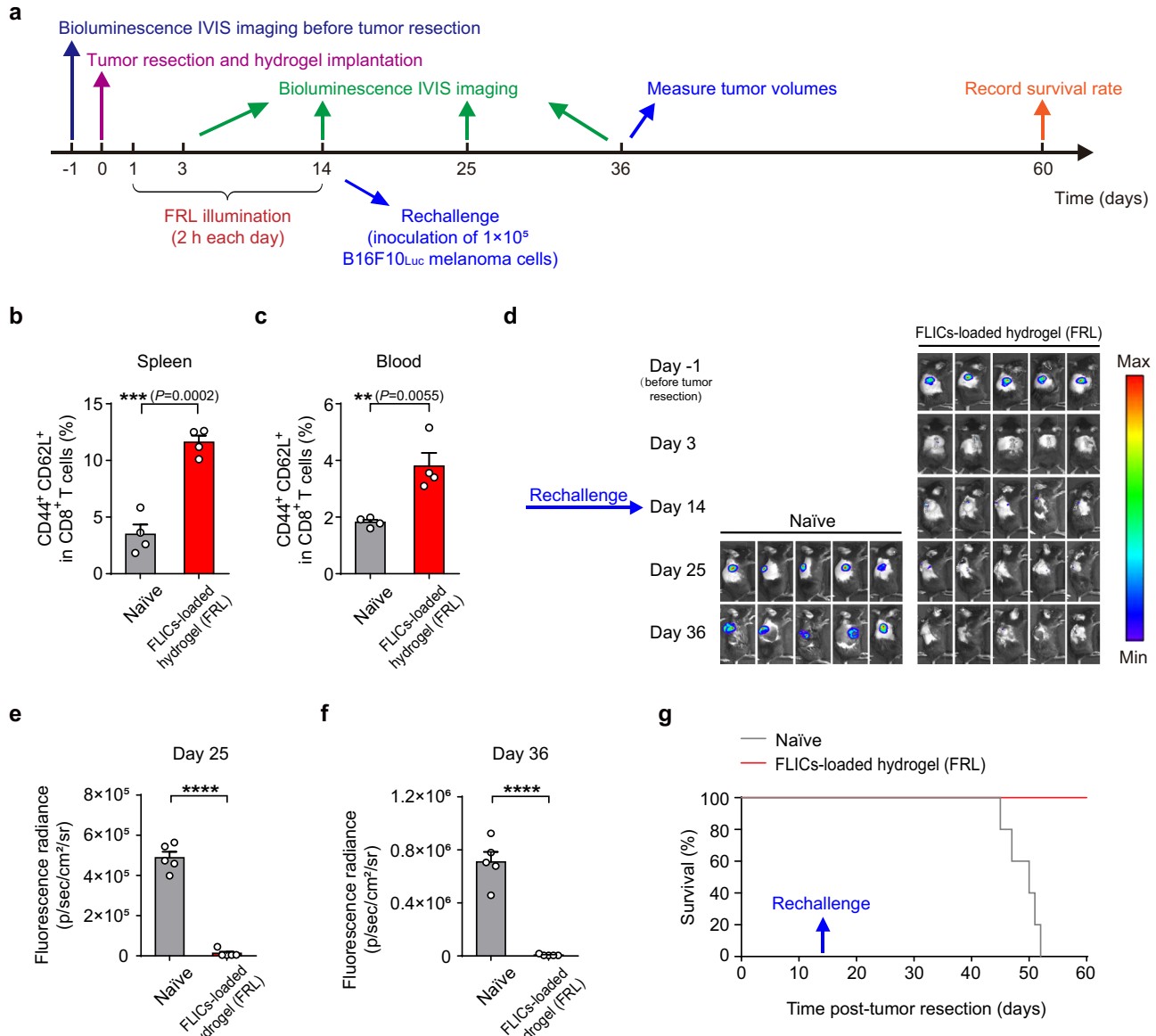

**Fig. 6 | Anti-tumor memory immune response induced by FLICs-loaded hydrogel implants in C57BL/6J mice. a** Schematic illustrating the experimental procedure and the time schedule for evaluating anti-tumor memory immune response. Tumor rechallenge was performed via subcutaneous injection of B16F10 melanoma cells ($1 \times 10^5$) at the resection site on day 14 after tumor removal. **b** Ratios of central memory-like (CD44⁺CD62L⁺) CD8⁺ T cells were evaluated in the spleens of control (naive) and treated (FLICs-loaded hydrogel implants and FRL illumination) mice sacrificed on day 14. Quantification of CD44⁺CD62L⁺CD8⁺ T cells. **c** Ratios of central memory-like CD8⁺ T cells were evaluated in peripheral blood of control (naive) and treated (FLICs-loaded hydrogel implants and FRL illumination) mice on day 14 by flow cytometric analysis. Quantification of central memory-like CD8⁺ T cells. **d** In vivo bioluminescence imaging of tumor burden performed on day 25 and day 36 after tumor resection (i.e., 11 and 22 days after rechallenge). Five mice per group are shown. Quantification of tumor burden based on bioluminescence IVIS imaging data from **d** on day 25 (**e**) and day 36 (**f**). **g** Kaplan-Meier curve of survival following tumor rechallenge ($n = 5$ mice). Data in **b**, **c**, **e**, **f** are presented as the mean ± SEM (**b**, **c**, $n = 4$ mice; **e**, **f**, $n = 5$ mice). $P$ values were calculated by two-tailed unpaired $t$-test. \*\*$P < 0.01$, \*\*\*$P < 0.001$, \*\*\*\*$P < 0.0001$. Source data are provided as a Source Data file.

obviously an obstacle for further clinic application. An ideal solution is to develop an optogenetic switch with a simple design and rapid activation/deactivation features in response to light. Note that our recently reported red/far-red light-mediated and minimized ΔPhyA-based photoswitch (REDMAP) system was able to induce therapeutic expression in different animals under a much shorter illumination (minute scale)[37], which might provide a promising alternative tool for optogenetics-based cell therapy.

Beyond its potential for cancer therapy, the engineered living cell factory cells that we loaded into implants should be easily adaptable for remote control expression and secretion of other protein drugs including enzymes, peptide hormones, vaccines, and even antibodies for treating various diseases. For example, chassis cells (e.g., mesenchymal stem cells) loaded in scaffolds could be engineered to produce urate oxidase for treating gout[38], insulin or human glucagon-like peptide 1 for treating diabetes[39,40], cancer vaccines for stimulating immature dendritic cells (DCs) and naive T cells[41], erythropoietin for treating anemia[42], interferons for treating virus infection[43], and even monoclonal antibodies (e.g., anti-PDL-1)[44] or immune checkpoint inhibitors for the treatment of cancer[45,46]. These ideas advance progress towards the living cell factory concept and this robust platform can be harnessed for driving long-term and on-demand production of therapeutic outputs for treating different diseases in a traceless remote-controllable defined manner.

## Methods

### Ethical statement

All experiments involving animals were performed according to the protocol approved by the ECNU Animal Care and Use Committee and in direct accordance with the Ministry of Science and Technology of the People's Republic of China on Animal Care Guidelines. The protocol was approved by the ECNU Animal Care and Use Committee (protocol ID: m20200209). All mice were euthanized after the termination of the experiments.

### Plasmid construction

Design and construction details for all expression vectors are provided in Supplementary Table 1. Plasmids were cloned using Gibson assembly according to the manufacturer's instructions (MultiS One Step Cloning Kit; catalog no. C113-01; Vazyme Inc.). All cloned genetic components were confirmed by Sanger sequencing (Genewiz Inc.). The complementary DNAs (cDNAs) encoding IFN-β, TNF-α and IL-12 were chemically synthesized by the Genewiz Inc. (Suzhou, China).

All genetic components related to this paper are available with a material transfer agreement and can be requested from H.Y. (hfye@bio.ecnu.edu.cn).

### Cell culture and transfection

Telomerase-immortalized human mesenchymal stem cells (hMSC-TERT; SCRC-4000, ATCC) and FLICs (FRL-controlled immunomodulatory engineered cells) were cultured in Dulbecco's Modified Eagle Medium (DMEM; catalog no. C11995500BT; Gibco) with 10% (v/v) FBS (catalog no. FBSSA500-S; AusGeneX Inc.) and 1% (v/v) penicillin/streptomycin (catalog no. B540732-0010; Sangon Biotech Inc.). B16F10$_{Luc}$ (Luciferase tagged mouse B16F10 melanoma cancer cell line) and B16F10-OVA (ovalbumin) melanoma cells were purchased from Shanghai Sciencelight Biology Science & Technology Inc. and cultured in RPMI 1640 Medium (catalog no. 11875093; Gibco) with 10% (v/v) FBS and 1% (v/v) penicillin/streptomycin. The third or fourth passages of B16F10$_{Luc}$ cells were used for tumor-bearing experiments. All cell lines were incubated at 37 °C in a humidified atmosphere containing 5% $CO_2$ and regularly tested for the absence of *mycoplasma* and bacterial contamination.

hMSC-TERT cells were transfected with an optimized polyethyleneimine (PEI)-based protocol. Briefly, hMSC-TERT cells were seeded in a 24-well cell culture plate ($6 \times 10^4$ cells per well) 18 h before transfection and were subsequently co-transfected with corresponding plasmid mixtures for 6 h with 50 μL PEI and DNA mixture [PEI and DNA at a ratio of 3:1 (wt/wt)] (PEI, molecular weight 40,000, stock solution 1 mg/mL in ddH$_2$O; catalog no. 24765; Polysciences Inc.).

### Generation of stable cell lines

The stable cell line FLIC, transgenic for the constitutive FRL-inducible co-expression of mouse IFN-β, TNF-α, and IL-12 was constructed by co-transfecting hMSC-TERT cells with pYH88 (ITR-P$_{hCMV}$-BphS-P2A-YhjH-P2A-p65-VP64-BldD-P2A-mCherry-pA::P$_{mPGK}$-PuroR-pA-ITR), pYH428 (ITR-P$_{FRL}$-IFN-β-P2A-TNF-α-P2A-IL-12-P2A-EGFP-pA::P$_{mPGK}$-ZeoR-pA-ITR), and the Sleeping Beauty transposase expression vector P$_{hCMV}$-SB100X (P$_{hCMV}$-SB100X-pA) at a ratio of 15:10:2. After selection with 0.5 μg/mL puromycin (catalog no. A1113803; Life Technologies) and 100 μg/mL zeocin (catalog no. R25001; Life Technologies) for two weeks, the surviving population was picked for further cultivation and stimulated by FRL using a custom-designed $4 \times 6$ light-emitting diode (LED) array (1 mW/cm$^2$; 730 nm) 4 h per day for 2 days. Cytokine production in the culture supernatant was scored at 48 h after the first illumination. The monoclonal FLIC lines that showed high sensitivity to FRL were used for the following studies. All stable cell lines were regularly tested for the absence of *mycoplasma* and bacterial contamination.

### Cytokine detection

Cytokines (IFN-β, TNF-α, and IL-12) in the cell culture supernatant were measured using corresponding enzyme-linked immunosorbent assay (ELISA) kits according to the manufacturer's instructions (IFN-β, catalog no. 42400-2, PBL assay science Inc.; TNF-α, catalog no. 88-7324-86, Invitrogen; IL-12, catalog no. BMS616TEN, Invitrogen) and were quantified using a Synergy H1 hybrid multi-mode microplate reader with Gen5 software (version: 2.04). Cytokines in mouse plasma were quantified using a bead-assisted multiplex cytokine profiling kit according to the manufacturer's instructions (LEGENDplex™ Multiplex Assays; Biolegend Inc.). Briefly, for each reaction, 25 μL of plasma samples were diluted with 25 μL assay buffer prior to the test and were transferred to the sample wells. Then 25 μL of mixed beads were added to each sample well. The plate was sealed with a plate sealer and was placed on a plate shaker at 500 rpm for 2 h at room temperature. Then 200 μL of 1 × wash buffer was added to each well. After washing, 25 μL of biotinylated detection antibodies were added and each detection antibody binds to its specific analyte bound on the capture beads, thus forming capture bead-analyte-detection antibody sandwiches. Then, 25 μL of streptavidin-phycoerythrin (SA-PE) was subsequently added, the phycoerythrin (PE) signal fluorescence intensity is then quantified using a BD LSRFortessa™ Flow Cytometer (BD Biosciences) and the concentration of each cytokine was determined based on a known standard curve using the LEGENDplex™ data analysis software.

### FRL-controlled cytokine secretion from FLICs

FLICs ($6 \times 10^4$) were seeded in a 24-well cell culture plate. After the 18-h culture, the culture plate was placed below a custom-designed $4 \times 6$ LED array (730 nm; each LED was centered above a single well; Epistar Inc.) and illuminated for different time periods (0, 2, 4, 6 h) or light intensities (0, 0.2, 1, 5 mW/cm$^2$) once a day for two days by FRL. Cytokine production in the culture supernatant was quantified at 48 h after the first illumination.

### Preparation of different hydrogel implants

For the preparation of FLICs-loaded hydrogel implants, FLICs ($2.5 \times 10^6$) suspended in 60 μL Dulbecco's Modified Eagle Medium were gently mixed with 240 μL of VitroGel® 3D solution (catalog no. TWG001; the Well Bioscience) to make a FLICs-loaded hydrogel implant and then was transferred to a well of 48-well cell culture plate. Hydrogel matrix is crosslinked by ion (such as $Ca^{2+}$ or $Na^+$) interactions from the cell culture medium and then form soft hydrogel. Then, 600 μL cell culture medium was added to the soft hydrogel and incubated in an atmosphere of 5% $CO_2$ at 37 °C for 20 min to form a solid hydrogel implant.

For the preparation of the recombinant cytokine protein-loaded hydrogel implants, 1000 ng of IFN-β, 60 ng of TNF-α, and 200 ng of IL-12 were selected to dissolve in 60 μL PBS containing 5% trehalose and then added into 240 μL of VitroGel® 3D solution (catalog no. TWG001; the Well Bioscience) based on the total amount of IFN-β, TNF-α, and IL-12 over 15 days by testing the average concentration of the cytokines secreted from FLICs-loaded hydrogel implants every three days for 15 days. The hydrogel was allowed to cross-link for at least 20 min.

For the preparation of the hydrogel implants loaded with cells constitutively expressing cytokines, hMSC-TERT cells were transfected with an optimized polyethyleneimine (PEI)-based protocol. Briefly, hMSC-TERT cells were seeded in a 10-cm cell culture dish ($4 \times 10^6$ cells per dish) 18 h before transfection and were subsequently transfected with pYH500 (P$_{hCMV}$-IFN-β-P2A-TNF-α-P2A-IL-12-P2A-EGFP-pA, 25 μg) for 6 h with 1 mL PEI and DNA mixture [PEI (1 mg/mL) and DNA at a ratio of 3:1 (wt/wt)]. The transfected cells ($2.5 \times 10^6$) suspended in 60 μL DMEM were gently mixed with 240 μL of VitroGel® 3D solution (catalog no. TWG001) to make a hydrogel implant and then was transferred to a well of 48-well cell culture plate. Then, 600 μL cell culture medium

was added to each well and incubated in an atmosphere of 5% $CO_2$ at 37 °C for 20 min to form a solid hydrogel implant.

## FRL-controlled cytokine secretion from FLICs-loaded hydrogel implants

A FLICs-loaded hydrogel implant (containing $2.5 \times 10^6$ FLICs) was seeded in a well of 48-well cell culture plate. After 1 h, the culture plate was placed below a custom-designed $4 \times 6$ LED array and illuminated with different light intensities (0, 0.2, 1, 5 mW/cm$^2$) for 4 h or different time periods (0, 2, 4, 6 h) once a day for two days by FRL. Cytokine production in the culture supernatant was quantified at 48 h after the first illumination. To detect the long-term cytokine production performance in vitro, FLICs-loaded hydrogel implants were illuminated with FRL (1 mW/cm$^2$; 730 nm) for 4 h once a day for 30 days. Culture supernatants were collected to quantify IFN-β, TNF-α and IL-12 production, respectively, using corresponding ELISA kits every three days. The cell culture medium was refreshed every three days.

## Reversibility performance of cytokines (IFN-β, TNF-α, and IL-12) release mediated by the FLICs-loaded hydrogel implants

For in vitro study of the reversibility performance of cytokines release, each FLICs-loaded hydrogel implant (containing $2.5 \times 10^6$ FLICs) was seeded in a well of a 48-well cell culture plate containing 600 μL cell culture medium. The implant was illuminated with FRL (1 mW/cm$^2$; 730 nm) for 20 min (ON) or kept in the dark (OFF). Next, the implant was again illuminated for a reversal illumination condition at the beginning of each following 24-h cycle and the culture medium was exchanged every 24 h, and cytokines (IFN-β, TNF-α, and IL-12) production was quantified every 6 h for 72 h.

For in vivo study of the tunability of cytokines (IFN-β, TNF-α, and IL-12) release, mice given FLICs-loaded hydrogel implants were illuminated with FRL (1 mW/cm$^2$; 730 nm) for 2 h (ON) and without FRL for 22 h (OFF) every 24 h for 14 days. Cytokines (IFN-β, TNF-α, and IL-12) production were quantified every 6 h on day 1, 2, 7, 8, 13, 14 using LEGENDplex™ Multiplex Assay Kits.

## Recombinant cytokine protein injection

Mouse IFN-β (catalog no. 300-02BC; Peprotech Inc.), mouse TNF-α (catalog no. 315-01 A; Peprotech Inc.), and mouse IL-12 (catalog no. 210-12; Peprotech Inc.) were reconstituted in ddH$_2$O at 1 mg/mL, respectively, and then aliquoted the reconstituted solution or stored at −80 °C to minimize freeze-thaw cycles. Based on the average concentration of the cytokines secreted from FLICs-loaded hydrogel implants every three days for 15 days, 200 ng of IFN-β, 12 ng of TNF-α, and 40 ng of IL-12 were selected to mix with 100 μL PBS containing 5% trehalose before each experiment and subcutaneously injected at the tumor resection site of each mouse, once every three days over 15 days, five times in total.

## Melanoma mouse model

The female C57BL/6J wild-type mice (6-8 weeks old; ECNU Laboratory Animal Center) were kept in an animal house maintained at 22 ± 2 °C, with a 12-h light-dark cycle and free access to food and water. A total of $5 \times 10^5$ of luciferase-tagged B16F10 (B16F10$_{Luc}$) melanoma cells or B16F10-OVA (ovalbumin) melanoma cells were suspended in 100 μL sterile PBS and subcutaneously injected into the dorsum of the mice. The tumor burden was monitored by the bioluminescence signal of B16F10 melanoma cells.

## In vivo bioluminescence and imaging

Each mouse was intraperitoneally injected with 15 mg/mL D-luciferin (10 μL/g of body weight, catalog no. luc001; Shanghai Sciencelight Biology Science & Technology Inc.) in Dulbecco's PBS (DPBS), and anesthetized with 2% isoflurane before bioluminescence imaging. Ten minutes after luciferin injection, bioluminescence images of the mice

were obtained by using the IVIS Lumina II in vivo imaging system (Perkin Elmer, USA). Regions of interest were quantified as average radiance (p/sec/cm$^2$/sr) using Living Image® 4.3 software.

## FLICs-loaded hydrogel implants for preventing tumor recurrence

When tumor volume reached about 80 to 100 mm$^3$ (7 days for B16F10 tumor-bearing mice), the mice were anaesthetized using 2% isoflurane and the tumor was resected. Immediately after surgery, the hydrogel implants were placed in the resection site. The wounds were then closed using medical clips. The experiments were performed independently at least three times, and the experimenter was blinded to the groups of mice. For the tumor resection model to evaluate the functionality of the FLICs-loaded hydrogel implants, the mice were randomly divided into five groups (G1-G5) with six mice in each group. Mice were then implanted with empty hydrogel implants (G1), hMSC-TERT-loaded hydrogel implants (G2), FLICs-loaded hydrogel implants without FRL illumination (G4), FLICs-loaded hydrogel implants with FRL illumination (G5) and subcutaneously injected with recombinant cytokine (IFN-β, TNF-α, and IL-12) proteins at the resected tumor site (G3). Twelve hours after tumor resection, mice of G3 were injected with the recombinant cytokine mixture (200 ng of IFN-β, 12 ng of TNF-α, 40 ng of IL-12 per mouse) every three days over 15 days, and mice of G5 were illuminated with FRL (10 mW/cm$^2$; 730 nm) for 2 h once a day for 14 days. The tumor burden was monitored by the bioluminescence signal of B16F10$_{Luc}$ cells using an IVIS Lumina II in vivo imaging system (Perkin Elmer, USA). The tumor volumes were measured with a digital caliper and calculated by the following formula: Tumor volume = [length of tumor × (width of tumor)$^2$]/2. Animals were euthanized with carbon dioxide asphyxiation when the volume of the tumor exceeded 2.5 cm$^3$.

## Cytokine production analysis in mouse plasma

Mouse peripheral blood was collected and transferred to ethylenediaminetetraacetic acid (EDTA) coated mini vacutainer tubes (BD Biosciences). After 10 min incubation, samples were centrifuged at $1000 \times g$ for 10 min at 4 °C. Plasma was collected and immediately detected by multiplexing laser bead-based immunoassays. The cytokine array assays were performed to quantify cytokine (IFN-β, TNF-α, IL-12, CXCL10, IFN-α, and IFN-γ) levels in the blood using LEGENDplex™ Multiplex Assay Kits (Biolegend Inc.) according to the manufacturer's instructions. For each reaction, 25 μL of plasma sample was diluted with 25 μL assay buffer prior to the test. All samples were run on a BD LSRFortessa™ Flow Cytometer (BD Biosciences) and the concentrations of six cytokines were determined based on a known standard curve using the LEGENDplex™ data analysis software.

## Isolation of peripheral blood lymphocytes

Peripheral blood samples were collected from mouse orbit and transferred to ethylenediaminetetraacetic acid (EDTA) coated mini vacutainer tubes (BD Biosciences), then immediately isolated in a centrifuge at $1000 \times g$ for 10 min and plasma was discarded. The blood sample was lysed using a 1× red blood cell (RBC) lysis buffer (catalog no. B54100-0100; Sangon Biotech Inc.) for 15 min. Subsequently, the samples were centrifuged at $400 \times g$ for 10 min to remove the lysed RBCs in the supernatant and the pellets containing lymphocytes were washed using 1× PBS and resuspended in 1 mL 1× PBS.

## Isolation of spleen lymphocytes

Spleen cell suspensions were obtained in sterile conditions by grinding the tissue through a 70 μm cell strainer (strong nylon mesh with 70-micron pores, Thermo Fisher Scientific). The collected cells were treated with RBC lysis buffer (catalog no. B54100-0100; Sangon Biotech Inc.) for 15 min. Subsequently, the samples were centrifuged at $650 \times g$ for 10 min to remove the lysed RBCs in the supernatant and the

pellet containing the spleen lymphocytes was washed using 1× PBS and resuspended in 1 mL 1× PBS.

## Flow cytometry

To evaluate which immune cells are required to confer the observed anti-tumor effect, specific cell subsets (NK cells, CD69+ NK cells, CD11b+CD27+ NK cells, CD8+ T cells, CD69+CD8+ T cells) were stained with fluorescence-labelled antibodies. The antibodies used for marking NK cells were Alexa Fluor®−700 anti-mouse CD45 (catalog no. 103127, clone 30-F11), Brilliant Violet 421TM-anti-mouse CD3 (catalog no. 100228, clone 17A2), FITC-anti-mouse NK-1.1 (catalog no. 108705, clone PK136). The antibodies used for marking CD69+ or CD11b+CD27+ NK cells were Alexa Fluor®−700 anti-mouse CD45 (catalog no. 103127, clone 30-F11), Brilliant Violet 421TM-anti-mouse CD3 (catalog no. 100228, clone 17A2), FITC-anti-mouse NK-1.1 (catalog no. 108705, clone PK136), PE/Cyanine7-anti-mouse CD69 (catalog no. 104511, clone H1.2F3), or APC/Cyanine7-anti-mouse CD11b (catalog no. 101226, clone M1/70), PerCP/Cyanine5.5-anti-mouse CD27 (catalog no. 124213, clone LG.3A10). The antibodies used for marking CD8+ T cells were Alexa Fluor®−700 anti-mouse CD45 (catalog no. 103127, clone 30-F11), Brilliant Violet 421TM-anti-mouse CD3 (catalog no. 100228, clone 17A2), FITC-anti-mouse CD8α (catalog no. 100705, clone 53-6.7). The antibodies used for marking CD69+CD8+ T cells were Alexa Fluor®−700 anti-mouse CD45 (catalog no. 103127, clone 30-F11), Brilliant Violet 421TM-anti-mouse CD3 (catalog no. 100228, clone 17A2), FITC-anti-mouse CD8α (catalog no. 100705, clone 53-6.7) and PE/Cyanine7-anti-mouse CD69 (catalog no. 104511, clone H1.2F3). To evaluate which immune cells are required to confer the anti-tumor memory immune response, specific cell subsets (CD44+CD62L+CD8+ T cells) were stained with fluorescence-labelled antibodies. The antibodies used for marking CD44+CD62L+CD8+ T cells were Alexa Fluor®−700 anti-mouse CD45 (catalog no. 103127, clone 30-F11), Brilliant Violet 421TM-anti-mouse CD3 (catalog no. 100228, clone 17A2), FITC-anti-mouse CD8α (catalog no. 100705, clone 53-6.7), PE/Cyanine7-anti-mouse CD44 (catalog no. 103029, clone IM7), and APC-anti-mouse CD62L (catalog no. 104411, clone MEL-14). All antibodies were purchased from Biolegend, and were diluted 1:200. The stained cells were measured on a BD LSRFortessaTM Flow Cytometer (BD Biosciences). A minimum of 5000 events per plot were collected and analyzed using the FlowJo V10 software. The numbers presented in the flow cytometry analysis images are percentage based.

## Analysis of intracellular IFN-γ production

To carry out IFN-γ production analysis, splenocytes were collected from mouse spleen and incubated with phorbol myristate acetate (PMA, 50 μg/mL) and ionomycin (50 μg/mL) for 5 h at 37 °C in T-cell culture medium (RPMI 1640 supplemented with 10% FBS and 1% penicillin-streptomycin). Next, the cells were washed in PBS buffer and stained with Alexa Fluor®−700 anti-mouse CD45 (catalog no. 103127, clone 30-F11), Brilliant Violet 421TM-anti-mouse CD3 (catalog no. 100228, clone 17A2), and FITC-anti-mouse CD8α (catalog no. 100705, clone 53-6.7) for 30 min on ice. All antibodies were obtained from Biolegend (San Diego, CA), and were diluted 1:200. Cells were then permeabilized using an intracellular fixation and permeabilization solution (fixation/permeabilization concentrate (catalog no. 00-5123-43; eBioscience) mixed with fixation/permeabilization diluent (catalog no. 00-5223-56; eBioscience) at 1:3 for 30 min at 4 °C. Finally, cells were stained with PE-anti-mouse IFN-γ (catalog no. 505808, clone XMG1.2) for 30 min on ice, washed in PBS buffer and analyzed using a BD LSRFortessaTM Flow Cytometer (BD Biosciences).

## Depletion of NK cells and CD8+ T cells

To evaluate which immune cells were required to confer the observed antitumor effect, one day before tumor resection and hydrogel implantation, NK cells and CD8+ T cells in mice were depleted by intraperitoneal injection of depleting antibodies (200 μg/mouse) every 3 days for five times in total. The antibodies used for depletion were anti-mouse NK-1.1 (catalog no. 108760; Biolegend) and anti-mouse CD8α (catalog no. 100764; Biolegend).

## Analysis of antigen-specific CD8+ T cell populations

C57BL/6J mice were subcutaneously inoculated with $5 \times 10^5$ B16F10-OVA (ovalbumin) melanoma cells. When the tumor volume reached about 80 to 100 mm³, the tumor was resected and the hydrogel implants were placed in the resection site. The mice were randomly divided into three groups (G1-G3) with six mice in each group. Mice were then implanted with FLICs-loaded hydrogel implants with (G3) or without FRL illumination (G2). Control mice were not given FRL illumination (dark) or mice were not given a post-resection implant (G1). On day 7 post-resection, spleen lymphocytes were isolated and treated with OVA$_{257-264}$ peptide (SIINFEKL, 10 μg/mL) for 12 h at 37 °C in T-cell culture medium [RPMI 1640 supplemented with 10% FBS, 1% penicillin-streptomycin and 2 μM monensin (catalog no. 50501ES03, Yeasen Inc.)]. After washing with PBS, cells were stained with Alexa Fluor®−700 anti-mouse CD45 (catalog no. 103127, clone 30-F11), Brilliant Violet 421TM-anti-mouse CD3 (catalog no. 100228, clone 17A2), and FITC-anti-mouse CD8α (catalog no. 100705, clone 53-6.7) for 30 min on ice, and then permeabilized with intracellular fixation and permeabilization solution. Finally, cells were stained with PE-anti-mouse IFN-γ (catalog no. 505808, clone XMG1.2) for 30 min on ice, washed with PBS and analyzed using a BD LSRFortessaTM Flow Cytometer (BD Biosciences). All antibodies were purchased from Biolegend, and were diluted 1:200.

## Systemic anti-tumor immune response mediated by FLICs-loaded hydrogel implants

For the distant tumor model to evaluate the systemic immune responses mediated by FLICs-loaded hydrogel implants, mice were subcutaneously (s.c.) inoculated with B16F10$_{Luc}$ melanoma cells ($5 \times 10^5$) in both their right and left flanks. When the tumor size reached about 80-100 mm³, tumors on both flanks were resected. The mouse tumor resection models were randomly assigned to three groups with five mice in each group. The mice were implanted with FLICs-loaded hydrogel implants only on the right side (without or with FRL illumination). Control mice were not given any implant, or without FRL. The B16F10 melanoma tumor was monitored using an IVIS Lumina II in vivo imaging system on day 3, 10, 20 and day 30 after tumor resection.

## CD8α labeling of paraffin tumor tissue sections

The tumors were collected from the mice and fixed with 4% paraformaldehyde for 24 h. Samples were dehydrated through a series of graded ethanol baths and then infiltrated with paraffin. Then, 5-μm-thick sections were obtained using a rotary microtome (Leica RM2235, Manual Rotary Microtome), rehydrated and permeabilized in 0.5% Triton X-100 (dissolved in PBS; catalog no. 9002-93-1; Sigma) for 20 min. Nonspecific staining between the primary antibodies and the tissue samples was blocked by incubating sections in the block buffer (1% fetal bovine serum in PBS) for 1 h at room temperature. After incubating with the anti-mouse CD8α antibody (1:500; catalog no. ab217344; Abcam) overnight at 4 °C, the slides were washed three times for 15 min each time in PBS and then incubated with the Alexa Fluor 488 goat anti-rabbit immunoglobulin G antibody (1:1000; catalog no. ab150077; Abcam) for 40 min at room temperature. After washing three times with PBS, the slides were incubated with DAPI solutions (5 μg/ml; catalog no. Cl002; Beyotime Inc.) for 10 min at room temperature. The slides were further washed three times with PBS, mounted in the antifade mounting media, and imaged by a fluorescence microscope (DMI8; Leica) equipped with an Olympus digital camera (Olympus DP71; Olympus). CD8α-positive cytomembrane was stained green, and all nuclei were stained blue.

## Anti-tumor memory immune response induced by FLICs-loaded hydrogel implants

For tumor rechallenge experiments to assess memory immune response induced by FLICs-loaded hydrogel implants, mice treated with FLICs-loaded hydrogel implants (under FRL illumination) were rechallenged with $1 \times 10^5$ B16F10$_{Luc}$ melanoma cells at the original tumor site on day 14 after tumor removal to develop new tumors and control wild-type (naive) mice were not given any implant, or without FRL. The tumor burden was monitored using an IVIS Lumina II in vivo imaging system on day 3, 14, 25, 36 after tumor resection. In addition, control wild-type (naive) mice and treated (FLICs-loaded hydrogel implants with FRL illumination) mice were sacrificed on day 14 post-resection and central memory-like cells from the spleens and peripheral blood of these mice were collected and analyzed using flow cytometry.

## Statistical analysis

All in vitro data are expressed as the mean ± SD of three independent experiments ($n = 3$). For the animal experiments, each treatment group consisted of randomly selected mice ($n = 4$ to 6). The results are expressed as means ± SEM. Statistical significance was analyzed by the Student's $t$ test. Survival was plotted using Kaplan–Meier curves. Neither animals nor samples were excluded from the study. Statistical analyses were performed using GraphPad Prism software version 6.0.

## Reporting summary

Further information on research design is available in the Nature Research Reporting Summary linked to this article.

## Data availability

All data associated with this study are present in the Article, Supplementary Information. Source data are provided with this paper.

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

## Acknowledgements
This work was financially supported by grants from the National Natural Science Foundation of China (no. 31971346 and 31861143016), the National Key R&D Program of China, Synthetic Biology Research (no. 2019YFA0904500), the Science and Technology Commission of Shanghai Municipality (no. 22N31900300 and 18JC1411000), and the Fundamental Research Funds for the Central Universities to H.Y. This work was also partially supported by the National Key R&D Program of China (no. 2019YFA0110802) and the National Natural Science Foundation of China (no. 32171414 and 31870861) to M.W., by the National Natural Science Foundation of China (81771306), the Science and Technology Commission of Shanghai Municipality (21S11906200, 201409002900) to W.J. and by the Natural Science Foundation of Shanghai (grant no. 18ZR1436000) to F.C. We also thank the ECNU Multifunctional Platform for Innovation (011) for supporting the mice experiments and the Instruments Sharing Platform of the School of Life Sciences, ECNU.

## Author contributions
H.Y. conceived the project. H.Y., Y.Y., X.W., W.J., and M.W. designed the experiments, analyzed the results, and wrote the manuscript. Y.Y., X.W., M.W., W.L., Y.Z., L.Z. X.Z. and Z.H. performed the experimental work. Y.Y., X.W., M.W., W.L, F.C., W.J., Q.Z., and H.Y. designed, analyzed and interpreted the experiments. All authors edited and approved the manuscript.

## Competing interests
The authors declare no competing interests.
