## [Peer Review File · Nature Communications]

Optogenetic-controlled immunotherapeutic designer cells for post-surgical cancer immunotherapyREVIEWER COMMENTS

Reviewer #1 (Remarks to the Author): with expertise in optogenetics

This is an innovative study highlighting the development of far-red light (FRL)-controllable immunomodulatory engineered cells (FLICs). The authors repurposed a technology previously pioneered by the same group to enable inducible production of cytokines, including IFN- β , TNF- α , and IL-12, when coupled with hydrogel upon LED-based FRL illumination. The authors have demonstrated the FRL-inducible cytokine secretion both in vitro and in vivo, showcasing their potential translational values in the prevention of tumor recurrence. The article is very well-written, and the major conclusions are well supported by the high quality rigorous data. Even though the level of conceptual innovation is not outstanding, the technical innovation is unquestionable with very clever design and well-designed experiments. Hence, the overall enthusiasm for this study is quite high. But some revisions are needed prior to consideration of publication, as outlined below.

1. The author did not explain well why they need to employ hydrogel in this study, is it used to manage the bio-distribution of cytokines or for the sake of easier optogenetic control? This begs some other related questions: How did the experiment outcome differ from simply using FLIC cells (which were not embedded into the hydrogel)? How long does the hydrogel stay in the body of the experimental mice? Is it necessary to re-inject new hydrogel-loaded FLIC cells in mice once the hydrogel is degraded? If so, what are the benefits of this system? Is hydrogel causing any negative effects in experimental mice? It's convincing that the system can be controlled to release cytokine in response to light. In terms of expense and effort, however, intermittent injections of cytokines into mice or humans would be far more practical than the sophisticated FLICs approach. Please describe the benefits of this system and draw a side-by-side comparison with the state of the art in cytokine therapy routine adopted in the clinical setting. These points should be discussed throughout the manuscript (best if some of the points can be validated experimentally).
2. Will the ectopically expressed BphS elicit host immunogenicity? This seems to be unaddressed throughout the manuscript. In addition, the FRL-induced production of c-di-GMP might also activate host cell innate immune signaling. This caveat should be taken into account when interpreting the observed phenotypes.
3. Fig. 3: Why did the control group, which received subcutaneous injections of IFN-, TNF-, and IL-12 (Group 3, Fig.3b), fail to achieve the substantial results as Group 5? Please explain this discrepancy.
4. The authors claimed that the system is tailored for spatiotemporal control, but later on, the authors concluded in Fig. 5b and the explanation in the manuscript (page 7, line 188 to 191) that "local tumor recurrence (i.e., on the right side) was significantly inhibited by FLICs-loaded hydrogel implants (with FRL), and also observed that the extent of tumor recurrence was significantly reduced on the left side." As a result, this does not seem to be a case of tight spatial control. Please elaborate on the limitation of the technology to make a fair conclusion.
5. Clone 8 appears to be the best clone in terms of producing three cytokines (judging from Fig. S2). Why is it not used for further characterization? Please provide the rationale for clone selection.
6. For better visual effects, please present the mice images in larger panels.

Reviewer #2 (Remarks to the Author): with expertise in cancer immunology/immunotherapy

Cancer recurrence after surgical resection remains a challenge in cancer therapy. In this manuscript, the authors developed a optogenetic-controlled immunotherapy, FLICs, which achieved the precise control production of immunomodulatory cytokines to against tumor recurrence after resection. In addition, authors also showed a inhibition of distant tumor growth and no obvious tumor growth after re-challenge. This optogenetic-based local delivery system combined with surgery is interesting and could have impact on current practice. However, the immune related mechanism should to be further clarified.

Major issues:

As neoadjuvant therapy, many immunotherapies have been used before surgery. Applying

immunotherapy in early stage of tumor progression could active more antigen-specific T cells, which can transfer to non-tumor position and further protect against the recurrence and metastasis of tumors. However, authors applied cytokine treatment after surgery and still showed an effective protection.

1. What's the mechanism of FLICs-based immunotherapy after surgery? Which immune cells are essential, DC, NK, CD4, or CD8
2. In the manuscript, authors only showed the percentage of total CD8 T cells. How about the antigen-specific CD8 T cells?
3. The expression of CD69 only show the activation of T cell or NK, not function. Which cytokines or molecules are essential, such as IFN γ ?

Minor issues:

1. Fig.3. why use 200ng IFN β , 12ng TNF α , 40 ng IL12? what are the bases of these dose?
2. Fig.4. M and O should be total CD8 T cells, not cytotoxic T cells.

Reviewer #3 (Remarks to the Author): with expertise in immunotherapy and immuno-engineering

Cancer immunotherapy is an important and timely topic of research, with many advances in the past decade. This manuscript aims to address the side effects that are common with many immunotherapies via the local delivery via transferred cells of cytokines. The authors developed a quite complex approach to cytokine delivery in this work, and this is likely to face significant barriers to translation. Genetically modifying cells, adaptively transferring and then the need to trigger cytokine release creates a quite complex system, and it is not clear such complexity is actually needed. Further, the studies do not support many of the authors conclusions.

Specific concerns:

The animal studies are missing key controls: (1) a hydrogel or other sustained release system (e.g., minipump) continuously releasing the same cytokines, as versus using cells triggered to release. Sustained release systems are quite straightforward to create and use, and no data is provided by the authors to demonstrate an advantage over this much more simple approach to accomplish the same goal. The authors provide a control with repeated injections of dissolved cytokines, but not a simple sustained release system. (2) another control of cells constitutively secreting the cytokines. The authors do not demonstrate the need for the main capability of the system (ability to turn on and off cytokine release) as versus simply having a continuous low, level of release from the cells. To achieve cytokine release, the authors illuminate animals for 2 hr, for 14 days in a row in their studies. This would be very unattractive from a clinical perspective, as it would be difficult to similarly treat patients. The authors do not explore how various triggering schemes impact the immune response.

The authors do not demonstrate light modulated secretion (turn on and off) in vivo, which is important to interpret the data in the manuscript.

The major motivation for development of the system here is to minimize side-effects, but the authors do not directly address in the manuscript. This is particularly relevant as the authors data indicates that substantial quantities of the presumably cell-secreted cytokines are present in the systemic circulation.

As tumors recur in patients at significant times post-resection, likely the key to prevent recurrence is to enable treatment at later time points. It is unclear if transferred cells will survive in vivo for any reasonable time frame, as versus ideal culture conditions in vitro – particularly in the context of human therapy where one would presumably need to transplant much higher numbers of cells than what is pursued here. In addition, the GTP levels may be low in what one would expect to be hypoxic cells in the gels in the body, making the mechanism of light-induced activation much less efficient.

In the data of Fig. 3 – very low mouse number was used in the survival study (n=6), and in many studies of the manuscript. Were the key studies repeated to ensure reproducibility?

Point-by-point responses to referees' comments:

We would like to thank all of the referees for their highly constructive comments.

As we hope you will agree, the careful revision process we have undertaken has substantially improved both the scientific rigor and impact of our study. We present point-by-point responses to each of the referee comments (below). We therefore invite you to examine our responses, and we would again like to thank all of the referees for their ongoing work on our behalf.

REVIEWER COMMENTS

Reviewer #1 (Remarks to the Author): with expertise in optogenetics

This is an innovative study highlighting the development of far-red light (FRL)-controllable immunomodulatory engineered cells (FLICs). The authors repurposed a technology previously pioneered by the same group to enable inducible production of cytokines, including IFN- β , TNF- α , and IL-12, when coupled with hydrogel upon LED-based FRL illumination. The authors have demonstrated the FRL-inducible cytokine secretion both in vitro and in vivo, showcasing their potential translational values in the prevention of tumor recurrence. The article is very well-written, and the major conclusions are well supported by the high quality rigorous data. Even though the level of conceptual innovation is not outstanding, the technical innovation is unquestionable with very clever design and well-designed experiments. Hence, the overall enthusiasm for this study is quite high. But some revisions are needed prior to consideration of publication, as outlined below.

We appreciate the positive comments and constructive suggestions. We preformed new experiments and addressed all comments as detailed below.

1. The author did not explain well why they need to employ hydrogel in this study, is it used to manage the bio-distribution of cytokines or for the sake of easier optogenetic control? This begs some other related questions: How did the experiment outcome differ from simply using FLIC cells (which were not embedded into the hydrogel)? How long does the hydrogel stay in the body of the experimental mice? Is it necessary to re-inject new hydrogel-loaded FLIC cells in mice once the hydrogel is degraded? If so, what are the benefits of this system? Is hydrogel causing any negative effects in experimental mice? It's convincing that the system can be controlled to release cytokine in response to light. In terms of expense and effort, however, intermittent injections of cytokines into mice or humans would be far more practical than the sophisticated FLICs approach. Please describe the benefits of this system and draw a side-by-side comparison with the state of the art in cytokine therapy routine adopted in the clinical setting. These points should be discussed throughout the manuscript (best if some of the points can be validated experimentally).

Thanks for the comments. To protect the implanted cells from the host immune system while simultaneously allowing free diffusion of oxygen, ions, and secreted proteins, the far-red light-controlled immunomodulatory engineered cells (FLICs) were encapsulated within hydrogels in this study.

We conducted new experiments to study the degradation of the empty hydrogel implants and FLICs-loaded hydrogel implants in mice. Our new data showed that hydrogels could remain in mice for at least 8 weeks without observed degradation (**new Figure S5**).

Furthermore, the hydrogel implants loaded with FLICs could confer strong inhibition against tumor recurrence under FRL illumination (10 mW/cm^2 ; 730 nm) for 2 h each day for 14 days after the hydrogel scaffolds harboring FLIC were placed in the tumor resection site. Thus, we think it is not necessary to re-inject new FLICs-loaded hydrogel implants into mice.

To evaluate whether there were negative effects of the hydrogel implants in the experimental mice, we have studied the potential toxicity of the FLICs-loaded hydrogel implants at fourteen days after placement of the hydrogel implants. Routine blood test and serum biochemistry analyses showed that no obvious differences in these blood phenotypes were detected between controls (untreated healthy mice) and animals given implants and FRL illumination (**Figure S7**), indicating that the placement of the FLICs-loaded hydrogel implants at resected tumor sites does not induce obvious systemic adverse effects. Previous studies also reported that commercially available polysaccharide-based biocompatible hydrogel scaffolds do not cause negative effects *in vivo* (Powell, K., *Science*, 2017; Ogino, T., et al., *Elife*, 2021; Ali, A., et al., *J. Agric. Food Chem.*, 2018).

Actually, cytokine injection will offer a more practical route of administration. However, there are some challenges associated with cytokines. For example, the systemic administration of cytokines is not efficient due to their short half-life *in vivo* and poor bioavailability at the target site. This, in turn, requires repeated injections, resulting in side effects and more treatment costs (Choi, S. H., et al., *Clin. Cancer Res.*, 2017). Compared with cytokine injection, our optogenetic-controlled “living cell factory” could drive long-term and on-demand production of therapeutic outputs for treating diseases in a traceless remote-controllable defined manner and prevent excessive immune responses and their fatal side effects, simply by using a beam of light.

Moreover, we compared FLICs-loaded hydrogel implants with direct subcutaneous injection (five times in total) of the recombinant cytokine proteins to the resected tumor region and found the FLICs-

loaded hydrogel implants (with FRL illumination) were superior to recombinant cytokine protein injection in terms of recurrent tumor volume (**Figure 3**). These results suggest that the engineered “living cell factory” cells can produce cytokines at therapeutic concentrations by controlling light irradiation periods and intensities, and might open the door for further clinical development of safe cytokine immunotherapies. We have provided these points in the revised manuscript. Please see line 149-156.

2. Will the ectopically expressed BphS elicit host immunogenicity? This seems to be unaddressed throughout the manuscript. In addition, the FRL-induced production of c-di-GMP might also activate host cell innate immune signaling. This caveat should be taken into account when interpreting the observed phenotypes.

Thanks for noting this. In fact, at the beginning of this work, we performed an experiment to evaluate whether the ectopically expressed BphS elicits host immunogenicity or c-di-GMP activates the endogenous human STING pathway. We transfected the plasmids carrying the light receptor BphS and the FRL-dependent transactivator (p65-VP64-BldD) or only BphS into hMSC-TERT cells. Indeed, we found that IFN- α and IFN- β were induced after illumination. However, the very lower level (0.5-2 pg/mL) of IFN- α and IFN- β production was not enough to activate host cell innate immune signaling.

3. Fig. 3: Why did the control group, which received subcutaneous injections of IFN-, TNF-, and IL-12 (Group 3, Fig.3b), fail to achieve the substantial results as Group 5? Please explain this discrepancy.

Thank you for this question. The main reason is that subcutaneous injections of cytokines is less efficient for cancer immunotherapy due to the low stability and short half-life of cytokines. This phenomenon was also observed in other studies (Park et al., *Sci. Transl. Med.*, 2018; Choi, S. H., et al., *Clin. Cancer Res.*, 2017). Compared with direct subcutaneous injection, our optogenetic-controlled “living cell factory” was able to drive long-term sustainable and on-demand production of therapeutic cytokines by controlling light irradiation periods and intensities. Therefore, the antitumor therapeutic effects of Group 5 were superior to Group 3.

4. The authors claimed that the system is tailored for spatiotemporal control, but later on , the authors

concluded in Fig. 5b and the explanation in the manuscript (page 7, line 188 to 191) that “local tumor recurrence (i.e., on the right side) was significantly inhibited by FLICs-loaded hydrogel implants (with FRL), and also observed that the extent of tumor recurrence was significantly reduced on the left side.” As a result, this does not seem to be a case of tight spatial control. Please elaborate on the limitation of the technology to make a fair conclusion.

Thank you for your comment. The experiments of Figure 5 were only designed to investigate the FLICs-loaded hydrogel implant-mediated anti-tumor immune responses. They were not designed to show the spatiotemporal transgene activation performance. Compared with chemical-based transgene induction systems, this optogenetic-controlled system can achieve remote and traceless modulation of cellular activities in a non-invasive way. The data in Figure 5 demonstrate that FLICs-loaded hydrogel implants on the right side not only conferred inhibition of local tumor recurrence but also reduced the extent of tumor recurrence on the left side and substantially outperformed soluble immunotherapeutic cytokine administration. These data suggest that the FLICs-loaded hydrogel implants with illumination can induce antigen-specific CD8⁺ T cells responses and antitumor activity.

To facilitate the translational capability of optogenetic-controlled post-surgical cancer immunotherapy, the optogenetic system should be more sensitive in response to light, with a wide dynamic range, and should be relatively simple. Our group recently reported the red/far-red light-mediated and minimized Δ PhyA-based photoswitch (REDMAP) system with characterization of extreme sensitivity to light: as little as 0.2 mW/cm² red light, delivered for only 1 s was sufficient to induce ~50% of the maximum transcriptional activation (Zhou, Y., et al., *Nat. Biotechnol.*, 2022). We think the REDMAP system can be applied for optogenetic-controlled post-surgical cancer immunotherapy. We now have provided this point in the Discussion section in the revised manuscript.

5. Clone 8 appears to be the best clone in terms of producing three cytokines (judging from Fig. S2). Why is it not used for further characterization? Please provide the rationale for clone selection.

Thank you for this question. Due to the background of clone no. 8, the FRL-inducible production of three cytokines was 145-, 110-, and 50-fold induction compared to dark control cells. However, clone no. 16 exhibited FRL-inducible cytokines production with very low background and had 264-, 225-, and 276-fold induction compared to dark control cells. Therefore, clone no. 16 was chosen for further animal experiments. We have labeled the relative induction fold changes in the revised Figure S2.

6. For better visual effects, please present the mice images in larger panels.

We have provided the magnified images in the revised supplementary materials (page 16-20).

Reviewer #2 (Remarks to the Author): with expertise in cancer immunology/immunotherapy

Cancer recurrence after surgical resection remains a challenge in cancer therapy. In this manuscript, the authors developed a optogenetic-controlled immunotherapy, FLICs, which achieved the precise control production of immunomodulatory cytokines to against tumor recurrence after resection. In addition, authors also showed a inhibition of distant tumor growth and no obvious tumor growth after re-challenge. This optogenetic-based local delivery system combined with surgery is interesting and could have impact on current practice. However, the immune related mechanism should to be further clarified.

Thank you for your comments. We have very carefully considered your comments and performed new experiments to address these concerns, and the details of these experiments will be expanded below.

Major issues:

As neoadjuvant therapy, many immunotherapies have been used before surgery. Applying immunotherapy in early stage of tumor progression could active more antigen-specific T cells, which can transfer to non-tumor position and further protect against the recurrence and metastasis of tumors. However, authors applied cytokine treatment after surgery and still showed an effective protection.

Thank you for your comments. Indeed, for cancer patients, many immunotherapies have been used before surgery. Although it is generally thought that earlier treatment with immunotherapy would be beneficial, side effects such as delaying the timing of curative treatment, increasing treatment toxicity, and confounding accurate pathological staging after systemic administration have limited such studies to date. At present, surgical resection is the main treatment option for solid tumors. Moreover, surgery not only reduced primary resistance to immunotherapy by removing cancer cell–intrinsic mechanisms of resistance but also improved anti-tumor immune responses by removing cancer cell–extrinsic factors that promote primary and adaptive resistance (Sharma et al., *Cell*, 2017).

Several studies have worked with mouse tumor resection models and demonstrated that biomaterial-based local delivery systems could prevent tumor recurrence after primary resection by extending the release of diverse immune agonists (e.g., TLR7/8, STING) or antibodies (e.g., anti-CD47) (Park, C.G. et al., *Sci. Transl. Med.*, 2018; Chen, Q. et al. *Nat. Nanotechnol.* 2019). According to the literature, we developed far-red light-controlled immunomodulatory engineered cells (FLICs) and then embedded them within a hydrogel scaffold at resection sites. FLICs enabled the precise optogenetic control of cytokines release (IFN- β , TNF- α , and IL-12) upon LED-based FRL illumination to activate both the innate and adaptive immune systems, thereby preventing tumor recurrence.

In addition, the previous study (Krishnamoorthy et al. *J Natl Cancer Inst.*, 2021; Joshua I Gray et al.

Immunology, 2018) also demonstrated that neoadjuvant immunotherapy could improve antitumor efficacy by increasing the number and maintenance of tumor-specific CD8⁺ T cells, while the antigen was not required for maintenance of the central memory T cells. This outcome may partially explain the efficacy that we observed in the postoperative setting.

We now have provided this information in the Discussion section in the revised manuscript.

1. What's the mechanism of FLICs-based immunotherapy after surgery? Which immune cells are essential, DC, NK, CD4, or CD8

Thank you for your questions and comments. In order to explore the mechanism of FLICs-based immunotherapy after surgery, we have performed new experiments addressing the depletion of a specific lineage by direct administration of anti-marker antibodies. Firstly, we systematically depleted the different immune-lineages in the B16F10 melanoma cells using anti-mouse NK-1.1 for removing NK cells and anti-mouse CD8 α for removing CD8⁺ CTLs (200 μ g antibodies per mouse, once every three days over 15 days, five times in total, intraperitoneal injection). Then, the growth of recurrent tumors was monitored in the mice with or without anti-mouse NK-1.1 or anti-mouse CD8 α treatment on post-resection days 3, 10, 17, and 24 (based on the bioluminescence signals of B16F10_{Luc} cells), and mouse survival was assessed at the pre-determined end-point of 60 days. The results showed that depletion of NK cells and CD8⁺ T cells significantly attenuated the antitumor effects suggesting that NK cells and CD8⁺ T cells might play a crucial role in antitumor activity (**new Figure 4 l-m**). We now have provided the detailed description in the revised manuscript.

2. In the manuscript, authors only showed the percentage of total CD8 T cells. How about the antigen-specific CD8 T cells?

It is interesting to explore this question. We have performed new experiments to assess whether the inhibition of tumor recurrence upon implantation of FLICs-loaded hydrogel implants was associated

with antigen-specific CD8⁺ T cells. The mice were inoculated with 5×10⁵ B16F10-OVA (ovalbumin) melanoma cells followed by resection of tumors and randomly divided into three groups (G1-G3) with six mice in each group. Mice without a post-resection implant (G1, untreated control) or mice with FLICs-loaded hydrogel implants without FRL illumination (G2) were used as control. Mice in G3 were given FLICs-loaded hydrogel implants with FRL illumination (G3). Seven days after tumor surgical resection, T cell response was determined by *ex vivo* re-stimulation of splenocytes with OVA₂₅₇₋₂₆₄ peptide and CD8⁺ T cells producing IFN-γ were measured. As shown in **New Figure S11**, FLICs-loaded hydrogel implants under FRL illumination induced higher proportions of OVA₂₅₇₋₂₆₄ peptide-specific IFN-γ⁺CD8⁺ T cells from the total CD8⁺ T cell population compared with other control groups. These results indicate that post-resection installation of our FLICs-loaded hydrogel implants induced antigen-specific CD8⁺ T cells against tumor recurrence.

We have provided detailed information in the revised manuscript. Please see page 8, lines 210-219 and page 18-19, lines 519-535.

3. The expression of CD69 only show the activation of T cell or NK, not function. Which cytokines or molecules are essential, such as IFNγ?

Thank you very much for this constructive comment. We have performed new experiments to assess production of IFN-γ, an important mediator of anti-tumor immunity. *Ex vivo* stimulation with PMA/ionomycin increased the number of IFN-γ-producing CD8⁺ T cells in spleen cells from mice given the FLICs-loaded hydrogel implants (with FRL) (**new Figure 4k**).

We also have provided these points in the revised manuscript. Please see page 7-8, lines 199-202 and

page 17-18, lines 498-510.

Minor issues:

1. Fig.3. why use 200ng IFN β , 12ng TNF α , 40 ng IL12? what are the bases of these dose?

FLICs-loaded hydrogel implants were seeded into a 48-well cell culture plate and illuminated with FRL (1 mW/cm²; 730 nm) for 4 h once a day for 15 days. Culture supernatants were collected to quantify IFN- β , TNF- α , and IL-12 production using corresponding ELISA kits every three days (the cell culture medium was refreshed every three days). The average concentration of the cytokines tested every three days for 15 days was chosen for the subcutaneous injection amount of the recombinant proteins at the tumor resection site of each mouse.

We have provided the related information in the Methods section in the revised manuscript. Please see page 14.

2. Fig.4. M and O should be total CD8 T cells, not cytotoxic T cells.

Thanks for your careful review. We have revised as directed.

Reviewer #3 (Remarks to the Author): with expertise in immunotherapy and immuno-engineering

Cancer immunotherapy is an important and timely topic of research, with many advances in the past decade. This manuscript aims to address the side effects that are common with many immunotherapies via the local delivery via transferred cells of cytokines. The authors developed a quite complex approach to cytokine delivery in this work, and this is likely to face significant barriers to translation. Genetically modifying cells, adaptively transferring and then the need to trigger cytokine release creates a quite complex system, and it is not clear such complexity is actually needed. Further, the studies do not support many of the authors conclusions.

Thank you for your comments. Cell-based therapy is opening up the possibility of novel therapeutic solutions for diseases. In this study, we mainly focused on synthetic designer cell-based therapy that is considered a very promising technique in the next generation of medicine (Fischbach, M. A., et al., *Sci. Transl. Med.*, 2013; Esensten, J. H., et al., *Annu. Rev. Pathol.*, 2017; McNerney, M.P., et al., *Nat. Rev. Genet.*, 2021).

As for the complexity issue of our system, we have to take the liberty of opposing the view that such a quite complex approach to cytokine delivery is likely to face significant barriers to translation. It

must be noted that all the synthetic biology-inspired genetically engineered therapeutic cells are engineered with multiple genetic elements which enable biochemical linking of certain signal stimuli to initiate a cascade of biochemical reactions leading to a specific output. Cell-based therapeutic technology has already demonstrated its superiority over conventional drugs for treating some difficult-to-treat diseases. An example of their clinical potential is provided by CAR-T cells, which are already in clinical use to cure cancers.

Challenges remain in engineering of cells for therapeutic purposes. It is already clear that therapeutic designer cells will play a pivotal role in next-generation medicine with the rapid progression of synthetic biology.

Our system has shown some advantages as follows:

- (1) These optogenetic designer cells enable sustainable release of multiple immunomodulatory cytokines compared to traditional material-based local delivery systems.
- (2) Our system enables optogenetic control of cytokines production based on controllable LED-based FRL light irradiation periods and intensities avoiding excessive activation of the immune response.
- (3) The optogenetic designer cells can be easily shut down in the event of therapeutic-induced toxicity simply by switching off light.
- (4) The engineered “living cell factory” cells loaded into implants could be easily adaptable for remote control expression and secretion of other therapeutic drugs.

Our study demonstrates that this type of optogenetic immunotherapy can open the door for further clinical development of safe cytokine immunotherapies. This intrabody implantation/infusion of such engineered cells could allow continuous on-demand production and delivery of therapeutic molecules. Moreover, FLIC cells can be turned off by an external light inducer in the event of therapeutic-induced toxicity. Such an “emergency stop button” would offer a new layer of safety, especially for cell implants intended for long-term therapeutic purposes.

We believe current rapid advances in synthetic biology and related research areas will provide new strategies to overcome the various difficulties, given the enormous advantages that can be expected from fully personalized medicine.

Specific concerns:

The animal studies are missing key controls: (1) a hydrogel or other sustained release system (e.g., minipump) continuously releasing the same cytokines, as versus using cells triggered to release. Sustained release systems are quite straightforward to create and use, and no data is provided by the authors to demonstrate an advantage over this much more simple approach to accomplish the same goal. The authors provide a control with repeated injections of dissolved cytokines, but not a simple sustained release system. (2) another control of cells constitutively secreting the cytokines. The authors do not demonstrate the need for the main capability of the system (ability to turn on and off cytokine

release) as versus simply having a continuous low, level of release from the cells.

Thank you for your constructive suggestions. We have performed additional new control experiments on mice implanted with hydrogel scaffolds loaded with three recombinant cytokines (IFN- β , TNF- α , IL-12) or hydrogel scaffolds loaded with cells constitutively expressing the cytokines (IFN- β , TNF- α , IL-12). The new data (**new Figure S6**) showed that the hydrogel implants loaded with FLICs (FRL illumination) conferred stronger inhibition against tumor recurrence and significantly prolonged mouse survival rates compared with other control groups.

We speculate that differences in tumor recurrence and mouse survival rates between G2 and G4 were mainly due to the stability and releasing dynamics of cytokines. Cytokines have a short half-life and need to be delivered by drug delivery systems such as hydrogel scaffolds to extend their release rate. However, the amount of cytokine proteins released from the hydrogel scaffolds gradually decreased over time, which is a common phenomenon observed in a previously reported study (Chun Gwon Park, et al., *Sci. Transl. Med.*, 2018), while our cell-based therapies can achieve controllable production of therapeutic proteins at the required times, since the therapeutic proteins are genetically encoded.

In addition, multiplexing laser bead-based immunoassays of peripheral blood samples showed that mice given the FLICs-loaded hydrogel implants and FRL illumination had increased levels of IFN- β , TNF- α , IL-12, CXCL10, IFN- α , and IFN- γ (**Figure S8**). Of note, mice given the hydrogel implants loaded with cells constitutively secreting the cytokines showed the highest levels of IL-6 due to overstimulated immune responses among all the groups (**new Figure S9**). These findings support that

FLICs-loaded hydrogel implants trigger the expected IFN- β , TNF- α , and IL-12 secretion at therapeutic concentrations without causing obvious systemic toxicity, which open the door for further clinical development of safe cytokine immunotherapies.

We have provided this information in the revised manuscript. Please see lines 184-189.

To achieve cytokine release, the authors illuminate animals for 2 hr, for 14 days in a row in their studies. This would be very unattractive from a clinical perspective, as it would be difficult to similarly treat patients. The authors do not explore how various triggering schemes impact the immune response.

Thank you for this comment. We think that our system provides the possibility of combining designer “living cell factories” with FRL induction as a new concept for the precise release of immunomodulatory cytokines, and is a promising approach to help protect against tumor recurrence post-resection. Moreover, our optogenetics-based cell therapy relies solely on light induction (2 h/day) and does not require repeated injections or oral medication, which reduces side effects and increased treatment costs.

To facilitate the translational capability of our system, the optogenetic system should possess greater sensitivity in response to light, with a wide dynamic range, and should be relatively simple. Note that our group recently reported the red/far-red light-mediated and minimized Δ PhyA-based photoswitch (REDMAP) system with characterization of extreme sensitivity to light: as little as 0.1 mW/cm² red light, delivered for only 1 s at 0.2 mW/cm², was sufficient to induce ~50% of the maximum transcriptional activation (Zhou, Y., et al., *Nat. Biotechnol.*, 2022). We think the REDMAP system should work well with our optogenetic-controlled post-surgical cancer immunotherapy for *in vivo* applications.

The authors do not demonstrate light modulated secretion (turn on and off) *in vivo*, which is important to interpret the data in the manuscript.

Thank you for your suggestion. It would be easy to do the light-controlled cytokines secretion (ON

and OFF) in mice to satisfy this concern. Unfortunately, we are in severe lockdown in Shanghai and have not been allowed to access our laboratory for almost three months. If this reviewer insists on this data, we can perform additional experiments to collect the data when the lab is open. Thank you for your understanding.

The major motivation for development of the system here is to minimize side-effects, but the authors do not directly address in the manuscript. This is particularly relevant as the authors data indicates that substantial quantities of the presumably cell-secreted cytokines are present in the systemic circulation.

Thank you for your comment. In our study, FLICs-loaded hydrogel implants and FRL illumination achieved therapeutic concentrations of the cytokines (IFN- β , TNF- α , IL-12) enabling an accurate tuning of the immune response based on controlling light irradiation periods and intensities. Moreover, routine blood tests and serum biochemistry analyses showed FRL-triggered cytokine production from FLICs-loaded hydrogel implants did not induce systemic adverse effects (**Figure S7**).

Mice given the hydrogel implants loaded with cells constitutively expressing cytokines (IFN- β , TNF- α , IL-12) showed the highest level of IL-6 production among all groups due to the overstimulated immune response (**new Figure S9**). We anticipate that the optogenetic immunotherapy might provide a new strategy for controllable cell-based secretion of cytokines for tumor immunotherapy while minimizing side-effects. We now have provided this new data in the revised manuscript. Please see page 7.

As tumors recur in patients at significant times post-resection, likely the key to prevent recurrence is to enable treatment at later time points. It is unclear if transferred cells will survive *in vivo* for any reasonable time frame, as versus ideal culture conditions *in vitro* – particularly in the context of human therapy where one would presumably need to transplant much higher numbers of cells than what is pursued here.

Thank you for your comments. We agree that this is a proof-of-concept study. More translational preclinical trials need to be performed prior to human clinical trials. As for the question of *in vivo* survival of the transferred cells, we conducted new experiments to study the degradation of the empty hydrogel implants and FLICs-loaded hydrogel implants in mice. Our new data showed that hydrogels could stay in mice for at least 8 weeks without any observed degradation (**new Figure S5**). As for the number of cells for human therapy, we anticipate more cells can be loaded into larger sizes of the hydrogel implants. There are several implantation strategies developed for cell implantation in human clinical trials. For example, micro-encapsulated pluripotent stem-cell-derived pancreatic cells were implanted into type 1 diabetes patients to control blood glucose homeostasis (Ramzy et al., *Cell Stem Cell*, 2021).

In addition, the GTP levels may be low in what one would expect to be hypoxic cells in the gels in the body, making the mechanism of light-induced activation much less efficient.

Thank you for this comment. The hydrogel we used here was able to protect the implanted cells from the host immune system while simultaneously allowing free diffusion of oxygen, ions, and secreted proteins. In our study, FLICs-loaded hydrogel implant scaffolds were placed in the tumor resection site and illuminated with FRL (10 mW/cm²; 730 nm) 2 h per day for 14 days. The data showed that the cytokine production levels under FRL illumination in serum remained high on day 14 (**Figure S8**), indicating light-induced activation was not affected. Moreover, FLICs-loaded hydrogel implants (under FRL illumination) were able to confer protection against the recurrence of both local and distant tumors after more than one month. Thus, we think that even if GTP may be low in the cells, it will not affect the activation of the promoter to drive cytokine expression *in vivo*.

In the data of Fig. 3 – very low mouse number was used in the survival study ($n=6$), and in many studies of the manuscript. Were the key studies repeated to ensure reproducibility?

Thank you for this comment. We repeated these animal experiments and obtained similar results. Because of animal ethics, six mice in each group were used. The statistical information is provided in each data panel. In fact, a number of studies also reported that 5-10 mice were used to evaluate anti-tumor effects and survival ratios *in vivo* (Qian Chen et al., *Nat. Nanotechnol.*, 2019; Feihu Wang et al., *Nat. Biomed. Eng.*, 2020).

REVIEWER COMMENTS

Reviewer #1 (Remarks to the Author):

The authors have addressed all the concerns by performing the recommended experiments and adding the necessary controls/annotations/discussions.

Reviewer #2 (Remarks to the Author):

The authors have addressed most concerns

Reviewer #3 (Remarks to the Author):

The authors have addressed a number of the previous concerns, but a number of important issues remain unresolved.

The authors now present data with two of the requested controls for the in vivo studies – gels containing cells that constitutively secrete cytokines, and gels releasing protein that was loaded into the gels. The specifics of these studies are quite unclear, as I could not find a description in the Methods or legend of the specific quantities of protein loaded into the gels, nor the release kinetics of the proteins. The quantity loaded should match that released from the cells, with the triggering schedule used for that condition, and the gels should also provide a sustained release of the proteins.

A key question related to the comparison to the control of gel-based protein delivery, and the direct cytokine injection condition is the quantity of protein used in these control conditions. While the former is not clear, the authors determined the quantity of cytokine used in the direct injection by measuring the cytokine amounts in cell culture medium of the cells – analyzing these quantities by assaying the medium every 3 days. However, cytokines typically have a very short half-life in cell culture, so the quantities measured in this manner likely only reflect a small fraction of the total cytokine secreted by the cells over the course of the 3 days. To ensure an appropriate comparison, the authors should analyze the secreted quantities over a much shorter time period, and then appropriately scale the protein quantity in the control conditions.

Along the lines of the comparison of constitutively secreted versus light-triggered cytokine secretion, the authors provide Suppl Fig. 9 demonstrating significantly increased IL-6 levels in mice with the constitutive secretion condition. What are the levels of the secreted cytokine? This data would be very relevant to understanding if a fair comparison was being made between the two conditions – for example, in the constitutive secretion situation, is vastly greater cytokine being delivered?

In terms of the question regarding the durability of this approach, relative to the clinical situation in which cytokine release may be required at times much after the initial tumor resection, the authors now indicate that the gels remain in vivo for at least 8 weeks. However, this does not address the question of whether the cells encapsulated in the gels remain viable and responsive for this time-frame. Encapsulated cells are often lost quite rapidly after transplantation.

The authors do not demonstrate the ability to turn cytokine secretion on and off in vivo, which is the fundamental advance they claim in the manuscript. The authors indicate performing this study is not possible due to COVID-related restrictions on laboratory research, but it is not clear how the other new data was then generated? In any case, this data is required to support the conclusions of the manuscript.

Point-by-point responses to referees' comments: We appreciate two reviewers' positive comments that they are satisfied with our previous revised manuscript. In addition, we have very carefully considered the third reviewer's new comments. As we hope you will agree, the careful second round revision process we have undertaken has improved the scientific rigor and quality of our study. We present point-by-point responses to the third reviewer's comments (below). We invite you to examine our responses and would like to once again thank all of the referees for their helpful comments that have significantly improved our manuscript.

Reviewer's Comments:

Reviewer #1 (Remarks to the Author):

The authors have addressed all the concerns by performing the recommended experiments and adding the necessary controls/annotations/discussions.

We appreciate this reviewer's positive comments and ongoing support of our work. Many thanks.

Reviewer #2 (Remarks to the Author):

The authors have addressed most concerns

We are happy that this reviewer is satisfied with our revision work. Thanks.

Reviewer #3 (Remarks to the Author):

The authors have addressed a number of the previous concerns, but a number of important issues remain unresolved.

The authors now present data with two of the requested controls for the in vivo studies – gels containing cells that constitutively secrete cytokines, and gels releasing protein that was loaded into the gels. The specifics of these studies are quite unclear, as I could not find a description in the Methods or legend of the specific quantities of protein loaded into the gels, nor the release kinetics of the proteins. The quantity loaded should match that released from the cells, with the triggering schedule used for that condition, and the gels should also provide a sustained release of the proteins.

Thanks for this comment pointing out the unclear description of quantities of protein loaded into the gels. We have now provided this information in the revised *Method* section. Please see lines 374-389, and below:

“For the preparation of the recombinant cytokine protein-loaded hydrogel implants, 1000 ng of IFN- β , 60 ng of TNF- α , and 200 ng of IL-12 were selected to dissolve in 60 μ L PBS containing 5% trehalose

and then added into 240 μL of VitroGel® 3D solution (catalog no. TWG001; the Well Bioscience) based on the total amount of IFN- β , TNF- α , and IL-12 over 15 days by testing the average concentration of the cytokines secreted from FLICs-loaded hydrogel implants every three days for 15 days. The hydrogel was allowed to cross-link for at least 20 min.

For the preparation of the hydrogel implants loaded with cells constitutively expressing cytokines, hMSC-TERT cells were transfected with an optimized polyethyleneimine (PEI)-based protocol. Briefly, hMSC-TERT cells were seeded in a 10-cm cell culture dish (4×10^6 cells per dish) 18 h before transfection and were subsequently transfected with pYH500 (P_{hCMV} -IFN- β -P2A-TNF- α -P2A-IL-12-P2A-EGFP-pA, 25 μg) for 6 h with 1 mL PEI and DNA mixture [PEI (1mg/mL) and DNA at a ratio of 3:1 (wt/wt)]. The transfected cells (2.5×10^6) suspended in 60 μL DMEM were gently mixed with 240 μL of VitroGel® 3D solution (catalog no. TWG001) to make a hydrogel implant and then was transferred to a well of 48-well cell culture plate. Then, 600 μL cell culture medium was added to each well and incubated in an atmosphere of 5% CO_2 at 37°C for 20 min to form a solid hydrogel implant.”

In addition, we further conducted new experiments to evaluate the release kinetics of each cytokine pre-loaded in the hydrogel. A hydrogel pre-loaded with the three cytokines (IFN- β , TNF- α , and IL-12) were immersed in 600 μL DMEM medium. 300 μL of medium was taken out and the same amount of fresh medium was added back once a day for 15 days. The amount of released cytokines was then measured using corresponding enzyme-linked immunosorbent assay (ELISA) kits according to the manufacturer’s instructions (IFN- β , catalog no. 42400-2, PBL Assay Science Inc.; TNF- α , catalog no. 88-7324-86, Invitrogen; IL-12, catalog no. BMS616TEN, Invitrogen). Our new data revealed that the release of three cytokines (IFN- β , TNF- α , and IL-12) started with an initial burst and then followed by significant decrease in the following 7-8 days, showing the similar release profile of the three cytokines and 98% of them were released within 8 days in DMEM medium (**please find the new data below**).

Compared to the release profile of the three cytokines loaded into the gels, our FLICs can release the three cytokines when exposed to 4 h FRL illumination per day over 30 days (**Supplementary Fig. 4a-c**). Moreover, FLICs-loaded hydrogel implants displayed tunable cytokines secretion kinetics in cells and mice. These results indicate that the FLICs as “living cell factory” are superior to the hydrogels pre-loaded with cytokines.

A key question related to the comparison to the control of gel-based protein delivery, and the direct cytokine injection condition is the quantity of protein used in these control conditions. While the former is not clear, the authors determined the quantity of cytokine used in the direct injection by measuring the cytokine amounts in cell culture medium of the cells – analyzing these quantities by assaying the medium every 3 days. However, cytokines typically have a very short half-life in cell culture, so the quantities measured in this manner likely only reflect a small fraction of the total cytokine secreted by the cells over the course of the 3 days. To ensure an appropriate comparison, the authors should analyze the secreted quantities over a much shorter time period, and then appropriately scale the protein quantity in the control conditions.

Thanks for the comments. We have performed a new experiment to assess the secreted quantities of the three cytokines once a day over 15 days using corresponding ELISA kits. Our new experimental data (see below) showed that the total amount of the three cytokines over 15 days is comparable to that tested every 3 days for 15 days reported in this study (Supplementary Figure 4), indicating that

it is reasonable to use the current dosage of the three cytokines in the cytokine injection group as a control.

Along the lines of the comparison of constitutively secreted versus light-triggered cytokine secretion, the authors provide Suppl Fig. 9 demonstrating significantly increased IL-6 levels in mice with the constitutive secretion condition. What are the levels of the secreted cytokine? This data would be very relevant to understanding if a fair comparison was being made between the two conditions – for example, in the constitutive secretion situation, is vastly greater cytokine being delivered?

Thanks for the comments. We have performed new experiments to evaluate the levels of the secreted cytokines (IFN- β , TNF- α , and IL-12) in the tumor resection model mice including mice given the hydrogel implants loaded with FLICs with or without FRL illumination, mice given cells constitutively expressing the cytokines or control mice that were not given a post-resection implant. On day 3 after

implantation, mouse plasma was collected to quantify IFN- β , TNF- α and IL-12 production using LEGENDplex™ Multiplex Assay Kits. The new data (**Supplementary Figure 10a-c, see below**) showed that much higher levels of IFN- β , TNF- α and IL-12 were expressed in mice given the hydrogel implants loaded with hMSC-TERT constitutively expressing cytokines than that of mice given the hydrogel implants loaded with FLICs with FRL illumination. The constitutively expressing cytokines showed the highest IL-6 expression levels (**Supplementary Figure 10d**) probably due to the overstimulated immune response.

In terms of the question regarding the durability of this approach, relative to the clinical situation in which cytokine release may be required at times much after the initial tumor resection, the authors now indicate that the gels remain *in vivo* for at least 8 weeks. However, this does not address the question of whether the cells encapsulated in the gels remain viable and responsive for this time-frame. Encapsulated cells are often lost quite rapidly after transplantation.

We sincerely thank for the comments, which provide very helpful suggestions for us to perform a rigorous study on the performance of hydrogel-based cell therapies. We agree that a detailed characterization of the temporal profile of both cytokine secretion and cell viabilities is essential to

achieving optimal *in vivo* applications.

As discussed with the editor, we will not perform additional experiments to further confirm the cell durability in the hydrogel for 8 weeks. The previous studies have demonstrated that cells loaded within the polysaccharide-based biocompatible hydrogel remained viable after implanting for 6-12 weeks *in vivo* (Shanbhag, S., et al., *Stem Cell Res. Ther.*, 2021; Li, R., et al., *Sci. Adv.*, 2019; Bian, L., et al., *Proc. Natl. Acad. Sci. U.S.A.*, 2013).

We hope this manuscript will report on the novel optical strategy for post-surgical cancer immunotherapy, which will set the stage for more in-depth and wider investigations. Indeed, we do plan to initiate a close collaboration with experts in the field of biomaterials in our department and work on the optimization and customization of hydrogel to further improve the performance of our optogenetic cell therapy.

The authors do not demonstrate the ability to turn cytokine secretion on and off *in vivo*, which is the fundamental advance they claim in the manuscript. The authors indicate performing this study is not possible due to COVID-related restrictions on laboratory research, but it is not clear how the other new data was then generated? In any case, this data is required to support the conclusions of the manuscript.

Thanks for the suggestion. We have conducted new experiments to investigate the tunability of the FLICs-loaded hydrogel implants in mice to see whether we can turn cytokine secretion on and off. In this study, mice given FLICs-loaded hydrogel implants were illuminated with FRL (1 mW/cm²; 730 nm) for 2 h (ON) and without FRL for 22 h (OFF) every 24 h for 14 days. Cytokines (IFN- β , TNF- α , and IL-12) production was quantified every 6 h on day 1, 2, 7, 8, 13, 14 using LEGENDplex™ Multiplex Assay Kits. The new data demonstrated that the FLICs-loaded hydrogel implants displayed tunable cytokines secretion in mice (**Supplementary Figure 9a-d, see below**).

Schematic representation of the experimental procedure and the time schedule used for evaluating cytokine secretion turned on and off in mice and the results are as follows:

REVIEWERS' COMMENTS

Reviewer #3 (Remarks to the Author):

The authors have adequately addressed the earlier concerns - the new data demonstrating on/off of secretion in vivo is particularly important for supporting the manuscript conclusions and is a valuable addition.

Point-by-point responses to referees' comments:

REVIEWERS' COMMENTS

Reviewer #3 (Remarks to the Author):

The authors have adequately addressed the earlier concerns - the new data demonstrating on/off of secretion in vivo is particularly important for supporting the manuscript conclusions and is a valuable addition.

We appreciate Reviewer #3' positive comments that he/she is satisfied with our revised manuscript.